# On the Minimax Regret for Contextual Linear Bandits and Multi-Armed Bandits with Expert Advice

**Shinji Ito**
The University of Tokyo and RIKEN
shinji@mist.i.u-tokyo.ac.jp

## Abstract

This paper examines two extensions of multi-armed bandit problems: multi-armed bandits with expert advice and contextual linear bandits. For the former problem, multi-armed bandits with expert advice, the previously known best upper and lower bounds have been $O(\sqrt{KT \log \frac{N}{K}})$ and $\Omega(\sqrt{KT \frac{\log N}{\log K}})$, respectively. Here, $K$, $N$, and $T$ represent the numbers of arms, experts, and rounds, respectively. We provide a lower bound of $\Omega(\sqrt{KT \log \frac{N}{K}})$ for the setup in which the player chooses an expert before observing the advices in each round. For the latter problem, contextual linear bandits, we provide an algorithm that achieves $O(\sqrt{dT \log(K \min\{1, \frac{S}{d}\})})$ together with a matching lower bound, where $d$ and $S$ represent the dimensionality of feature vectors and the size of the context space, respectively.

## 1 Introduction

This paper considers problems of multi-armed bandits with expert advice (MwE) [Auer et al., 2002, Kale, 2014], linear bandits [Dani et al., 2008, Bubeck et al., 2012, Cesa-Bianchi and Lugosi, 2012], and contextual linear bandits (CLB) [Chu et al., 2011, Abbasi-Yadkori et al., 2011, Neu and Olkhovskaya, 2020, Hanna et al., 2023, 2024].

One contribution of this paper is to present the minimax regret for MwE, which addresses the open question posed by Seldin and Lugosi [2016]. For MwE, the well-known EXP4 algorithm [Auer et al., 2002] achieves $O\left(\sqrt{KT \log N}\right)$-regret, where $K$, $T$ and $N$ represent the numbers of arms, rounds and experts. This bound is, however, not always optimal for certain parameter settings of $K$ and $N$. In fact, as discussed by Seldin and Lugosi [2016], when $N = K$, the problem is reduced to the standard $K$-armed bandit problem, for which the minimax regret is $\Theta(\sqrt{KT})$ [Audibert and Bubeck, 2009]. This means that the regret bound of EXP4 has a gap of an $O(\sqrt{\log K})$-factor when $N = K$. On the side of upper bounds, Kale [2014] addressed this issue by providing an algorithm achieving $O\left(\sqrt{KT \log_+ \frac{N}{K}}\right)$,[1] where we denote $\log_+ x := \max\left\{\log x, 1\right\}$. This is minimax optimal for the case of $N = O(K)$. However, the minimax optimal bound for arbitrary settings of $K$ and $N$ has been an open question. The best known lower bound $\Omega\left(\sqrt{KT \frac{\log N}{\log K}}\right)$ is shown by Seldin and Lugosi [2016], who conjectured that this lower bound is minimax optimal. This paper provides a solution to this open question by providing a lower bound of $\Omega\left(\sqrt{KT \log_+ \frac{N}{K}}\right)$, which, together with the upper

---

[1]Kale [2014] deals with more general settings of the multi-armed bandit with expert advice in which only a limited number of expert advice are accessible in each round.

38th Conference on Neural Information Processing Systems (NeurIPS 2024).

Table 1: Upper bounds ($O(\cdot)$) and lower bounds ($\Omega(\cdot)$) on regret for three problems. BwE: multi-armed bandit with expert advice, LB: linear bandit, CLB: contextual linear bandit. $K$: number of arms, $N$: number of experts, $d$: dimensionality of feature vectors, $S$: size of the context space.

| Setup | Reference | Bound | Parameter |
|-------|-----------|-------|-----------|
| BwE | [Auer et al., 2002] | $O\left(\sqrt{KT\log N}\right)$ | |
| | [Kale, 2014] | $O\left(\sqrt{KT\log_+ \frac{N}{K}}\right)$ | |
| | [Seldin and Lugosi, 2016] | $\Omega\left(\sqrt{KT\frac{\log N}{\log K}}\right)$ | $N \geq K$ |
| | **[This work]** | $\Omega\left(\sqrt{KT\log_+ \frac{N}{K}}\right)$ | $N \geq K$ |
| LB | [Bubeck et al., 2012] | $O\left(\sqrt{dT\log K}\right)$ | |
| | **[This work]** | $O\left(\sqrt{dT\log_+ \frac{K}{d}}\right)$ | |
| | [Auer et al., 2002] | $\Omega\left(\sqrt{dT}\right) = \Omega\left(\sqrt{dT\log_+ \frac{K}{d}}\right)$ | $K = d$ |
| | [Dani et al., 2008] | $\Omega\left(\sqrt{d^2T}\right) = \Omega\left(\sqrt{dT\log_+ \frac{K}{d}}\right)$ | $K = 2^d$ |
| CLB | [Liu et al., 2024] | $O\left(d\sqrt{T\log T}\right)$ | |
| | **[This work]** | $O\left(\sqrt{dT\log_+ \left(K\min\left\{1, \frac{S}{d}\right\}\right)}\right)$ | |
| | **[This work]** | $\Omega\left(\sqrt{dT\log_+ \left(K\min\left\{1, \frac{S}{d}\right\}\right)}\right)$ | $K \leq 2^d \leq K^S$ |

bound by Kale [2014], implies that the minimax regret is $\Theta\left(\sqrt{KT\log_+ \frac{N}{K}}\right)$. This lower bound is shown for the problem setting in which the player need to choose an expert before observing the advices in each round. As noted in Remark 1 below, though this problem setup is more challenging than the "classical" setting where the player can observe all expert advice before selecting an arm, almost all known existing algorithms, including those in Table 1, work for this setting.

Contextual linear bandit problems are a generalization of multi-armed bandit problems in which each arm $i \in [K] = \{1, 2, \ldots, K\}$ is linked to a feature vector $\phi(X_t, i) \in \mathbb{R}^d$ that depends on the context $X_t$ at the $t$-th round, and the expected loss suffered for choosing $i$ is expressed as a linear function of the feature vector: $\langle \theta_t, \phi(X_t, i) \rangle$. In particular, this paper considers the problem setting of the stochastic-context and adversarial-loss model [Neu and Olkhovskaya, 2020, Liu et al., 2024], i.e., problems with stochastic $X_t$ and adversarial $\theta_t$. We also assume that we are given access to the distribution of contexts, similarly to previous studies, such as those by Neu and Olkhovskaya [2020]. Linear bandit problems can be considered as special cases of contextual linear bandits in which the context space is a singleton.

Upper and lower bounds on regret for (contextual) linear bandits are shown in Table 1. For linear bandit problems, we show a regret upper bound of $O\left(\sqrt{dT\log_+(K/d)}\right)$, which is slightly better than the known best bounds of $O(\sqrt{dT\log K})$ by previous studies [Bubeck et al., 2012, Cesa-Bianchi and Lugosi, 2012, Dani et al., 2007]. One notable aspect of this novel bound is that it matches lower bounds for two important special cases of $K = 2^d$ [Dani et al., 2008, Theorem 3], and $K = d$ (equivalent to the standard $K$-armed bandits). However, specifying the tight minimax bounds for arbitrary values of $K$ remains an open problem.

For contextual linear bandits, this paper presents an upper bound of $O(\sqrt{dT\log_+(K\min\{1, S/d\})})$ together with a matching lower bound, where $S = |\mathcal{X}|$ is the cardinality of the context space. Note that the regret upper bound of $O(d\sqrt{T\log T})$ by Liu et al. [2024] applies to the more general setting of the adversarial-context and adversarial-loss model. We also note that the problem formulation of contextual linear bandit problems in some existing studies [Neu and Olkhovskaya, 2020, Kuroki et al., 2024, Olkhovskaya et al., 2024] is different from ours. As noted in Liu et al. [2024], however, this problem of different formulations can be reduced to our setting with dimension $d|\mathcal{A}|$, where $|\mathcal{A}|$ is the maximum number of actions in their settings. See [Liu et al., 2024] for a more detailed review of previous studies.

To show the regret lower bound for BwE, we follow the approach employed in the study [Chen et al., 2024] on the minimax regret for problems interpolating problems of (full-information) online learning with expert advice and multi-armed bandit. They have shown that, for the generalization of a $K$-armed bandit problem in which each arm is associated with $\nu$ different experts, the minimax regret is $\Theta\left(\sqrt{KT \log \nu}\right)$. Inspired by their construction of the problem instance, we consider a BwE instance in which $N' = \Theta(N/K)$ experts give advice for choosing one of two arms and there are independent $\Theta(K)$ copies of such structures. This yields a lower bound of $\Omega(\sqrt{KT \log N'}) = \Omega(\sqrt{KT \log(N/K)})$.

In the proof of regret upper bounds for (contextual) linear bandits, we develop an algorithm based on the follow-the-regularized-leader (FTRL) approach with a Tsallis entropy regularizer with a parameter $\alpha \in (0, 1)$: $\psi_\alpha(w) = -\frac{1}{1-\alpha} \sum_{i \in [K]} (w(i)^\alpha - w(i))$. This approach can be interpreted as a generalization of the EXP2 algorithm [Bubeck et al., 2012]. In fact, EXP2 can be regarded as the FTRL approach with Shannon entropy regularization, which coincides with the limit of $\alpha$-Tsallis entropy as $\alpha$ approaches 1. This paper shows that the regret bounds can be further improved by not fixing $\alpha$ to 1 but by adjusting it appropriately. Similar approaches have been used successfully in various online learning problems including multi-armed bandits [Audibert and Bubeck, 2009], bandits with expert advice [Kale, 2014], graph bandits [Eldowa et al., 2024], and sleeping bandits [Nguyen and Mehta, 2024]. In addition to FTRL with Tsallis entropy regularization, to achieve $O\left(\sqrt{KT \log_+\left(\frac{KS}{d}\right)}\right)$-regret in contextual linear bandits, we combine a novel technique of *context-dependent learning rate*. More precisely, we introduce learning rate parameters $\eta(X_t)$ that change depending on the observed context $X_t \in \mathcal{X}$. We show that tuning $\eta : \mathcal{X} \to \mathbb{R}_{>0}$ leads to improved regret bounds.

**Notation** For a natural number $n$, denote $[n] = \{1, 2, \ldots, n\}$. For two real vectors $x = (x(i))_{i \in [d]}, y = (y(i))_{i \in [d]} \in \mathbb{R}^d$, let $\langle x, y \rangle$ denote the inner product between $x$ and $y$, i.e., $\langle x, y \rangle = \sum_{i \in [d]} x(i)y(i)$. Let $\mathcal{P}(K) = \left\{w \in [0,1]^K \mid \sum_{i \in [K]} w(i) = 1\right\}$ denote the set of probability distributions over $[K]$. For any symmetric matrices $A$ and $B$, denote $A \succeq B$ if and only if $A - B$ is positive-semidefinite. Let $\text{tr}(A)$ denote the trace of square matrices $A$.

## 2 Multi-armed bandit with expert advice

### 2.1 Problem setting

In the problem of *multi-armed bandits with expert advice* (BwE), the player is given the number of arms $K$ and the number of experts $N$. In each round, each expert $j \in [N]$ select an advice $e_t(j) \in [K]$, and the player chooses an expert $J_t \in [N]$. Then, after the player observes the expert advice $(e_t(j))_{j \in [N]}$, the player pulls the arm $I_t = e_t(J_t) \in [K]$ and gets feedback of the suffered loss $\ell_t(I_t) \in [0, 1]$. The performance of the player is measured by means of regret $R_T$ defined as

$$R_T = \max_{j^* \in [N]} \mathbf{E}\left[\sum_{t=1}^{T} \ell_t(I_t) - \sum_{t=1}^{T} \ell_t(e_t(j^*))\right].$$

**Remark 1.** We consider the problem setting in which the player choose the expert $J_t$ before observing expert advice $(e_t(j))_{j \in [N]}$. This setting is more challenging compared to the one where the player can observe all expert advice before selecting an arm, because the available information is more limited. On the other hand, existing algorithms by Auer et al. [2002], Kale [2014] can be applied to this more challenging setting and achieve the regret upper bounds shown in Table 1. Therefore, we have decided to adopt this setting in this study.

**Best expert identification** The *best expert identification* (BEI) problem is a variant of BwE problem in which the player aims to identify the expert attaining the minimum value of expected loss from as few feedbacks as possible. We here assume that $(e_t, \ell_t)$ follows an identical distribution $\mathcal{D}$ independently for all $t = 1, 2, \ldots$. Note that, for $(e, \ell) \sim \mathcal{D}$, all elements of $e$ and $\ell$ may be dependent, and that we impose the independence assumption only between data at different $t$. For each expert $j \in [N]$ we define $\mu_j = \mathbf{E}_{(e,\ell) \sim \mathcal{D}}[\ell(e(j))]$ and let $j^* \in [K]$ denote the *best expert* in terms of the expectation, i.e., $j^* \in \arg\min_{j \in [N]} \mu_j$. In the BEI problem, the player repeats selecting

arms and obtaining feedback for an arbitrary number of times $T$, and then outputs the estimated optimal expert $J_T \in [K]$.

For a positive number $\varepsilon$, an expert $j$ is called an $\varepsilon$-*optimal expert* if $\mu_j < \mu_{j^*} + \varepsilon$. An algorithm $\mathcal{A}$ for BEI is called $(\varepsilon, \delta)$-*probably approximately correct* (PAC) if it outputs an $\varepsilon$-optimal expert with probability at least $(1 - \delta)$. For any algorithm $\mathcal{A}$ for BEI, let $\mathcal{T}(\mathcal{A}, \mathcal{D})$ denote the expected value of the number of rounds $T$ until $\mathcal{A}$ is terminated when it is applied to the problem instance associated with $\mathcal{D}$.

## 2.2 Reduction from BEI to BwE

This section shows that BEI problems can be reduced to BwE problems. More precisely, given an algorithm for BwE problems with a regret upper bound, we can construct an $(\varepsilon, \delta)$-PAC algorithm with a bounded number of queries, as follows:

**Lemma 1.** *Suppose that there exists an algorithm $\mathcal{A}$ for BwE such that the regret is bounded as $R_T \leq r(T)$ for every $T$. For an arbitrary $\varepsilon \in (0, 1)$, let $T^*$ be such that $T^* \geq \frac{2500 \cdot r(T^*)}{\varepsilon}$. Then, there exists an $(\varepsilon, 0.05)$-PAC algorithm $\mathcal{A}'$ for BEI such that $\mathcal{T}(\mathcal{A}, \mathcal{D}) \leq T^*$ for any $\mathcal{D}$.*

*Proof.* Consider the following algorithm $\mathcal{A}'$ for best-expert identification: (i) Run $\mathcal{A}$ for $T^*$ rounds and let $T_j$ be the number of rounds in which the expert $j$ is chosen, i.e., $T_j = |\{t \in [T^*] \mid J_t = j\}|$. (ii) Output $\hat{J} = j$ with probability $T_j/T^*$, i.e., $\hat{J}$ is chosen so that $\Pr[J = j] = T_j/T^*$ for all $j \in [N]$. Let us show that this is an $(\varepsilon, 0.05)$-PAC algorithm.

Let $\mathcal{N}_\varepsilon \subseteq [N]$ denote the set of $\varepsilon$-optimal experts and let $\mathcal{N}_\varepsilon^c = [N] \setminus \mathcal{N}_\varepsilon$. Let $T_\varepsilon = \sum_{j \in \mathcal{N}_\varepsilon} T_j$ denote the number of choosing $\varepsilon$-optimal experts. We then have $R_{T^*} \geq \mathbf{E}\left[\varepsilon \cdot \sum_{j \in \mathcal{N}_\varepsilon^c} T_j\right] = \varepsilon \mathbf{E}\left[(T^* - T_\varepsilon)\right]$ as we suffer per-round regret of at least $\varepsilon$ in expectation, in every round when an expert in $\mathcal{N}_\varepsilon^c$ is chosen. Hence, from Markov's inequality, we have

$$\Pr\left[T_\varepsilon \leq 0.99T^*\right] = \Pr\left[T^* - T_\varepsilon \geq 0.01T^*\right] \leq \frac{100}{T^*} \mathbf{E}\left[T^* - T_\varepsilon\right]$$
$$\leq \frac{100}{T^*} \cdot \frac{R_{T^*}}{\varepsilon} \leq \frac{100}{T^*} \cdot \frac{r(T^*)}{\varepsilon} \leq \frac{100}{2500} = 0.04,$$

where the last inequality follows from the assumption of $T^* \geq \frac{2500 \cdot r(T^*)}{\varepsilon}$. Hence, from the construction of algorithm $\mathcal{A}'$,

$$\Pr\left[\hat{J} \in \mathcal{N}_\varepsilon^c\right] = \sum_{j \in \mathcal{N}_\varepsilon^c} \mathbf{E}\left[\frac{T_j}{T^*}\right] = \mathbf{E}\left[\frac{T^* - T_\varepsilon}{T^*}\right]$$
$$\leq \Pr\left[T_\varepsilon \leq 0.99T^*\right] \cdot 1 + \Pr\left[T_\varepsilon > 0.99T^*\right] \cdot 0.01 \leq 0.04 + 0.01 = 0.05,$$

which means that $\mathcal{A}'$ is an $(\varepsilon, 0.05)$-PAC algorithm for BEI. $\qquad\square$

## 2.3 Construction of problem instance

For any $p \in [0, 1]$, let $\mathrm{Ber}(p)$ represent a Bernoulli distribution of parameter $p$, i.e., if $X \sim \mathrm{Ber}(p)$, then $\Pr[X = 1] = p$ and $\Pr[X = 0] = 1 - p$.

Without loss of generality, we consider the case that $N$ and $K$ can be expressed as $N = N'm + 1$ and $K = 2m + 1$ for some positive integers $N'$ and $m$. We denote the set of experts by $\mathcal{E} = \{0\} \cup \{(u, v)\}_{u \in [m], v \in [N']}$ and the set of arms by $\mathcal{A} = \{0\} \cup \{(u, b)\}_{u \in [m], b \in \{0,1\}}$. The expert advice $e_t$ is given so that $e_t(0) = 0$ and $e_t((u, v)) \in \{(u, 0), (u, 1)\}$ for all $u \in [m]$ and $v \in [N']$.

Fix $u^* \in [m]$, $v^* \in [N']$, and $\varepsilon \in [0, 1)$. We define distributions of $(e, \ell)$ as follows:

- $\mathcal{D}(\varepsilon)$: When $(e, \ell)$ follows $\mathcal{D}(\varepsilon)$, $\ell(0)$ follows $\mathrm{Ber}((1 - \varepsilon)/2)$. For each $u \in [m]$, $\ell((u, 0))$ and $\ell((u, 1))$ are given by $\ell((u, 0)) = b_u$ and $\ell((u, 1)) = 1 - b_u$, where $b_u$ follows $\mathrm{Ber}(1/2)$. The expert advice is given by $e((u, v)) = (u, b_{uv})$, where each $b_{uv}$ follows $\mathrm{Ber}(1/2)$ for $u \in [m]$ and $v \in [N']$. All elements of $\ell(0)$, $(b_u)_{u \in [m]}$ and $(b_{uv})_{u \in [m], v \in [N']}$ are independent.

- $\mathcal{D}(\varepsilon, u^*, v^*)$: When $(e, \ell)$ follows $\mathcal{D}(\varepsilon, u^*, v^*)$, $\ell$ and $e$ follows the same distribution as $\mathcal{D}(\varepsilon)$ except for $e((u^*, v^*))$. The value of $e((u^*, v^*))$ is given as $e((u^*, v^*)) = (u^*, b_{u^* v^*}) = (u^*, |\ell((u^*, 0)) - b'|)$, where $b'$ is a random variable that follows $\mathrm{Ber}(1/2 - \varepsilon)$ independently of the other randomness. In other words, the second element of $e((u^*, v^*))$ is chosen so that $\ell(e(u^*, v^*))$ follows $\mathrm{Ber}(1/2 - \varepsilon)$.

The probability distribution of $\mathcal{D}(\varepsilon, u^*, v^*)$ is constructed so that the marginal distribution of $e$ for $(e, \ell) \sim \mathcal{D}(\varepsilon, u^*, v^*)$ is same as that for $\mathcal{D}(\varepsilon)$. Indeed, the values of $e$ is determined by $(b_{uv})_{u \in [m], v \in [N']}$, and the marginal distribution of $b_{u^*, v^*}$ for $\mathcal{D}(\varepsilon, u^*, v^*)$ is a mixture distribution given as $\Pr\left[\ell((u^*, 0)) = 0\right] \cdot \mathrm{Ber}\left(\frac{1}{2} - \varepsilon\right) + \Pr\left[\ell((u^*, 0)) = 1\right] \cdot \mathrm{Ber}\left(\frac{1}{2} + \varepsilon\right) = \frac{1}{2}\mathrm{Ber}\left(\frac{1}{2} - \varepsilon\right) + \frac{1}{2}\mathrm{Ber}\left(\frac{1}{2} + \varepsilon\right) = \mathrm{Ber}\left(\frac{1}{2}\right)$, which is equivalent to distributions for $\mathcal{D}(\varepsilon)$. In addition, if $(e, \ell) \sim \mathcal{D}$ follows $\mathcal{D}(\varepsilon, u^*, v^*)$, the expected loss for choosing expert $(u, v)$ is given as

$$\mathop{\mathbf{E}}_{(e, \ell) \sim \mathcal{D}(\varepsilon, u^*, v^*)} \left[\ell(e(u, v))\right] = \frac{1}{2} - \varepsilon \cdot \mathbf{1}\left[(u, v) = (u^*, v^*)\right],$$

where $\mathbf{1}[\cdot]$ represents the indicator function, i.e., $\mathbf{1}[E] = 1$ if $E$ is true and $\mathbf{1}[E] = 0$ otherwise. This means that, for the problem instance of BEI associated with $\mathcal{D}(\varepsilon, u^*, v^*)$, the expert $(u^*, v^*)$ is the only $(\varepsilon/2)$-optimal action.

We denote $\mathcal{D}_T(\varepsilon) = (\mathcal{D}(\varepsilon))^T$. Let $\mathcal{D}_T(\varepsilon, u^*)$ denote the uniform mixture of $\{(\mathcal{D}(\varepsilon, u^*, v^*))^T\}_{v^* \in [N']}$: $\mathcal{D}_T(\varepsilon, u^*) = \frac{1}{N'} \sum_{v^* \in [N']} (\mathcal{D}(\varepsilon, u^*, v^*))^T$.

## 2.4 Lower bound for best-expert identification problems

We first consider the special case of $m = 1$ and provide an instance-specific lower bound for BEI associated with $\mathcal{D}(\varepsilon)$ and $\{\mathcal{D}(\varepsilon, 1, v^*)\}_{v^* \in N'}$.

**Lemma 2.** *Suppose $m = 1$ and let $\mathcal{A}$ be an $(\varepsilon/2, 0.05)$-PAC algorithm for BEI. We then have $\mathcal{T}(\mathcal{A}, \mathcal{D}(\varepsilon)) \geq \frac{1}{2} \left\lfloor \frac{\ln(N'/4)}{4\varepsilon^2} \right\rfloor =: \frac{T^*}{2}$.*

*Proof.* Let $T$ be the number of rounds the algorithm $\mathcal{A}$ proceeds. Let $E$ denote the event that the algorithm $\mathcal{A}$ terminate at a round before $T^* + 1$ (i.e., $T \leq T^*$) and output $\hat{J} = 0$. Then, from Pinsker's inequality, we have $\left|\Pr\left[E|((e_t, \ell_t))_{t=1}^{T^*} \sim \mathcal{D}_{T^*}(\varepsilon)\right] - \Pr\left[E|((e_t, \ell_t))_{t=1}^{T^*} \sim \mathcal{D}_{T^*}(\varepsilon, 1)\right]\right| \leq D_{\mathrm{KL}}\left(\mathcal{D}_{T^*}(\varepsilon, 1) || \mathcal{D}_{T^*}(\varepsilon)\right) \leq 0.25$, where the last inequality follows from Lemma 9 in the supplementary. As $\mathcal{A}$ is an $(\varepsilon/2, 0.05)$-PAC algorithm, we have $\Pr\left[E|((e_t, \ell_t))_{t=1}^{T^*} \sim \mathcal{D}_{T^*}(\varepsilon, 1)\right] \leq 0.05$. By applying the union bound, we obtain $1 - \Pr\left[E|((e_t, \ell_t))_{t=1}^{T^*} \sim \mathcal{D}_{T^*}(\varepsilon)\right] = \Pr\left[T > T^*|((e_t, \ell_t))_{t=1}^{T^*} \sim \mathcal{D}_{T^*}(\varepsilon)\right] + \Pr\left[T \leq T^*, \hat{J} \neq 0|((e_t, \ell_t))_{t=1}^{T^*} \sim \mathcal{D}_{T^*}(\varepsilon)\right] \leq \Pr\left[T > T^*|((e_t, \ell_t))_{t=1}^{T^*} \sim \mathcal{D}_{T^*}(\varepsilon)\right] + 0.05$. Combining these inequalities, we obtain $\Pr\left[T > T^*|((e_t, \ell_t))_{t=1}^{T^*} \sim \mathcal{D}_{T^*}(\varepsilon)\right] \geq 0.95 - \Pr\left[E|((e_t, \ell_t))_{t=1}^{T^*} \sim \mathcal{D}_{T^*}(\varepsilon)\right] \geq 0.95 - 0.25 - \Pr\left[E|((e_t, \ell_t))_{t=1}^{T^*} \sim \mathcal{D}_{T^*}(\varepsilon, 1)\right] \geq 0.95 - 0.25 - 0.05 = 0.65$, which implies that $\mathcal{T}(\mathcal{A}, \mathcal{D}(\varepsilon)) \geq \mathbf{E}\left[T \mid ((e_t, \ell_t))_{t=1}^{T^*} \sim \mathcal{D}_{T^*}(\varepsilon)\right] \geq 0.65 T^*$. This completes the proof. $\square$

We can obtain a lower bound for BEI for general $m$ by using Lemma 2.

**Theorem 1.** *Let $\mathcal{A}$ be an $(\varepsilon/2, 0.05)$-PAC algorithm for BEI. We then have $\mathcal{T}(\mathcal{A}, \mathcal{D}(\varepsilon)) \geq \frac{m}{2} \left\lfloor \frac{\ln(N'/4)}{4\varepsilon^2} \right\rfloor = \frac{mT^*}{2}$.*

## 2.5 Lower bound for multi-armed bandits with expert advice

We are now ready to provide a lower bound for BwE.

**Theorem 2.** *For any algorithm for BwE and for any sufficiently large $T > 0$, there exists a problem instance for which $R_T \geq C\sqrt{TK \log_+ \frac{N}{K}}$, where $C > 0$ is a universal constant.*

*Proof.* We fix $T$ and set $m = \lfloor (K-1)/2 \rfloor$ and $N' = \lfloor (N-1)/m \rfloor$. Fix $\varepsilon = \Theta \left( \sqrt{\frac{m \log N'}{T}} \right)$ so that $T < \frac{m}{2} \left\lfloor \frac{\ln(N'/4)}{4\varepsilon^2} \right\rfloor$. Assume that $R_T < C\sqrt{TK \log_+ \frac{N}{K}} := r(T) = O\left( \sqrt{Tm \log N'} \right)$, where the universal constant $C > 0$ is sufficiently small so that $\frac{2500 \cdot r(T)}{\varepsilon} \le T$ holds. Then, from Lemma 1, there exists an $(\varepsilon, 0.05)$-PAC BEI algorithm $\mathcal{A}$ achieving $\mathcal{T}(\mathcal{A}, \mathcal{D}) \le T$ for any $\mathcal{D}$. This contradicts Theorem 1, which implies that $R_T \ge C\sqrt{TK \log_+ \frac{N}{K}}$. $\square$

This lower bound, together with the upper bound presented by Kale [2014], means that the minimax regret for BwE is of $\Theta \left( \sqrt{KT \log_+ \frac{N}{K}} \right)$.

# 3 Contextual linear bandit

## 3.1 Problem setting

Before the game starts, the player is given a set $\mathcal{X}$ of contexts, a finite set $[K]$ of arms, a feature mapping $\phi : \mathcal{X} \times [K] \to \mathbb{R}^d$. In each round, the environment chooses loss vector $\theta_t \in \mathbb{R}^d$. Then a context $X_t \in \mathcal{X}$ is drawn from a fixed distribution $\mathcal{D}$, and is revealed to the player. The player chooses an action $I_t \in [K]$ and get feedback of $\ell_t \in [-1, 1]$, of which expectation is $\langle \theta_t, \phi(X_t, I_t) \rangle$. For any policy $\pi^* : \mathcal{X} \to [K]$, we define the regret by

$$R_T(\pi^*) = \mathbf{E} \left[ \sum_{t=1}^T \langle \theta_t, \phi(X_t, I_t) \rangle - \sum_{t=1}^T \langle \theta_t, \phi(X_t, \pi^*(X_t)) \rangle \right], \quad R_T = \sup_{\pi^* : \mathcal{X} \to [K]} R_T(\pi^*).$$

We assume that $|\langle \theta_t, \phi(x, i) \rangle| \le 1$ for all $x \in \mathcal{X}$ and $a \in [K]$. We also pose the following assumption regarding the distribution of the context.

**Assumption 1.** We assume that the set of contexts is a finite set $\mathcal{X} = [S]$ and that there exists $L \ge S$ such that the probability $g(x) := \Pr[X_t = x]$ is bounded from below as $g(x) \ge 1/L$ for all $x \in [S]$. Assume that the function $g$ is given.

For any *randomized policy* $p : \mathcal{X} \to \mathcal{P}(K)$, we denote

$$V(p) = \mathop{\mathbf{E}}_{X \sim \mathcal{D}, I \sim p(X)} \left[ \phi(X, I) \phi(X, I)^\top \right], \quad \lambda(p) = \sup_{i \in [K], x \in \mathcal{X}} \phi(x, i)^\top V(p)^{-1} \phi(x, i).$$

**Assumption 2.** We assume that there exists an *exploration policy* $p_0 : \mathcal{X} \to \mathcal{P}(K)$ such that $\lambda(p_0) < \infty$.

Let $\lambda_0 > 0$ denote (an upper bound of) the value of $\lambda(p_0)$. We may assume that $\lambda_0 \le Ld$ without loss of generality. In fact, if we set $p_0(x)$ to be a g-optimal design (see, e.g., [Lattimore and Szepesvári, 2020, Section 21.1]) for $\mathcal{Z}_x = \{\phi(x, i) \mid i \in [K]\}$, we then have $\phi(x, i)^\top V(p)^{-1} \phi(x, i) \le \frac{1}{g(x)} \phi(x, i)^\top V(p_0(x))^{-1} \phi(x, i) \le dL$ holds for all $x \in \mathcal{X}$ and $i \in [K]$.

## 3.2 Algorithm

The proposed algorithm is based on the framework follow-the-regularized-leader (FTRL) with Tsallis entropy regularization. For $\alpha \in (0, 1)$, define a regularization function $\psi : \mathcal{P}(K) \to \mathbb{R}$ by

$$\psi(w) = \frac{1}{\alpha} \sum_{i \in [K]} (w(i) - w(i)^\alpha) = \frac{1}{\alpha} \left( 1 - \sum_{i \in [K]} w(i)^\alpha \right). \tag{1}$$

Using this regularizer, we define a randomized policy $q_t : \mathcal{X} \to \mathcal{P}(K)$ on the basis of FTRL, and set an arm-selection policy $p_t$, as follows:

$$q_t(x) = \arg\min_{w \in \mathcal{P}(\mathcal{A})} \left\{ \sum_{i \in [K]} w(i) \sum_{s=1}^{t-1} \left\langle \hat{\theta}_s, \phi(x, i) \right\rangle + \frac{1}{\eta(x)} \psi(w) \right\}, \quad p_t = (1 - \gamma) q_t + \gamma p_0 \tag{2}$$

---

**Algorithm 1** Contextual linear bandit algorithm based on FTRL with Tsallis entropy

---

**Input:** Feature mapping $\phi$, learning rates $(\eta(x))_{x \in \mathcal{X}}$, exploration policy $p_0$, parameters $\gamma, \alpha$
**for** $t = 1, \ldots, T$ **do**
    Compute $p_t$ given by (1) and (2).
    Observe $X_t$, choose $I_t$ drawn from $p_t(X_t)$, and get feedback of $\ell_t$.
    Compute $\hat{\theta}_t$ defined by (3).
**end for**

---

where $\hat{\theta}_t$ is an unbiased estimator defined below, $\eta(x) > 0$ is a learning rate parameter that will be specified later, and $\gamma \in (0, 1/2)$ is a parameter that will be chosen depending on $\eta$. In each round $t$, we choose $I_t$ following $p_t(X_t)$ and get feedback of $\ell_t$. We then compute an unbiased estimator $\hat{\theta}_t$ of $\theta_t$ defined by

$$\hat{\theta}_t = \ell_t V(p_t)^{-1} \phi(X_t, I_t). \tag{3}$$

This is in fact an unbiased estimator as

$$
\mathop{\mathbf{E}}_{X_t \sim \mathcal{D}, I_t \sim p_t(X_t)} \left[ \hat{\theta}_t \right] = \mathop{\mathbf{E}}_{X_t \sim \mathcal{D}, I_t \sim p_t(X_t)} \left[ \ell_t V(p_t)^{-1} \phi(X_t, I_t) \right]
$$
$$
= \mathop{\mathbf{E}}_{X_t \sim \mathcal{D}, I_t \sim p_t(X_t)} \left[ V(p_t)^{-1} \phi(X_t, I_t) \phi(X_t, I_t)^\top \theta_t \right] = V(p_t)^{-1} V(p_t) \theta_t = \theta_t. \tag{4}
$$

The procedure of our proposed algorithm is summarized in Algorithm 1.

### 3.3 Regret upper bound

This section provides an upper bound on the regret for Algorithm 1. In the following, we use the symbol $\beta = 1 - \alpha \in (0, 1)$ for simplicity of notation. We will show a regret upper bound as follows:

**Theorem 3.** *Suppose that $1/2 \leq \alpha < 1$ and that $\gamma$ satisfies*

$$
\gamma = \lambda_0 \cdot \min \left\{ 8 \sup_{x \in \mathcal{X}} \eta(x), \ \sup_{x \in \mathcal{X}} \left( \frac{16\eta(x)}{g(x)^\beta} \right)^{1/\alpha} \right\} \leq \frac{1}{2}. \tag{5}
$$

*Then, the regret for Algorithm 1 is bounded as follows:*

$$
R_T(\pi^*) = O \left( T \left( \frac{1}{\beta} \min \left\{ d \cdot \sup_{x \in \mathcal{X}} \eta(x), \ d^\alpha \cdot \sup_{x \in \mathcal{X}} \frac{\eta(x)}{g(x)^\beta} \right\} + \gamma \right) + \frac{K^\beta - 1}{\alpha} \sum_{x \in \mathcal{X}} \frac{g(x)}{\eta(x)} \right).
$$

From this theorem, by tuning parameters $\alpha$ and $\eta(x)$ on the basis of $T, d, K$ and $g$, we obtain the following:

**Corollary 1.** *For sufficiently large $T$, Algorithm 1 achieves the following:*

- *By setting $\eta(x) = \eta' g(x)^\beta =$ with $\eta' = \Theta \left( \sqrt{\frac{\beta(KS)^\beta}{\alpha d^\alpha T}} \right)$ and $\beta = \Theta \left( 1/ \left( \log_+ \left( \frac{KS}{d} \right) \right) \right)$, we obtain $R_T = O \left( \sqrt{dT \log_+ \left( \frac{KS}{d} \right)} + \lambda_0 \sqrt{T^{1-\beta}} \right)$.*

- *By setting $\eta(x) = \eta = \Theta \left( \sqrt{\frac{\beta K^\beta}{\alpha d T}} \right)$ for all $x \in \mathcal{X}$ and $\beta = \Theta \left( \frac{1}{\log K} \right)$, we obtain $R_T = O \left( \sqrt{dT \log K} + \lambda_0 \min \left\{ \sqrt{\frac{T}{d \log K}}, L^{\frac{\beta}{\alpha}} \sqrt{T^{1-\beta}} \right\} \right)$.*

**Remark 2.** The proposed algorithm (Algorithm 1) can also work for the infinite context case, in which it enjoys the second regret upper bound in Corollary 1. In fact, this regret upper bound does not include $S = |\mathcal{X}|$ or $L$, and the value of $g(x)$ is not required to define $\eta(x)$ in showing this second upper bound. Hence, we can show this bound without the assumption that $\mathcal{X}$ is a finite set. In the infinite context case, however, further challenges regarding the computational complexity of the algorithm should be noted. For example, we need to compute $V(p_t)$ in the algorithm as it appears in the definition of $\hat{\theta}_t$ in (3), which tend to be computationally expensive, depending on the computational model and the setup of distributions.

To prove Theorem 3, we introduce some notations: Denote $\hat{\ell}_t(x) = \left( \left\langle \hat{\theta}_t, \phi(x, i) \right\rangle \right)_{i \in [K]} \in \mathbb{R}^K$, which is an unbiased estimator of $(\langle \theta_t, \phi(x, i) \rangle)_{i \in [K]}$. For any $w, w' \in \mathcal{P}(K)$, let $D(w, w')$ denote the Bregman divergence associated with $\psi$, i.e., $D(w, w') = \psi(w) - \psi(w') - \langle \nabla \psi(w'), w - w' \rangle$. From the standard analysis technique for FTRL (see, e.g., [Lattimore and Szepesvári, 2020, Chapter 28]) and the idea of *ghost sample* $X_0 \sim \mathcal{D}$ drawn independently from all other randomness (see, e.g., [Neu and Olkhovskaya, 2020]), we obtain the following upper bound:

**Lemma 3.** *For Algorithm 1, the regret is bounded as* $R_T \leq 2\gamma T + \frac{K^\beta - 1}{\alpha} \sum_{x \in \mathcal{X}} \frac{g(x)}{\eta(x)} + \mathbf{E} \left[ \sum_{t=1}^{T} \left( \left\langle \hat{\ell}_t(X_0), q_t(X_0) - q_{t+1}(X_0) \right\rangle - \frac{1}{\eta(X_0)} D(p_{t+1}(X_0), p_t(X_0)) \right) \right].$

We note that $X_0$ is a random variable that does not appear in the decision-making process or algorithms, but appear only in the analysis, and is defined to be independent of $X_1, X_2, \ldots, X_T$ (and therefore is independent of any other variables including $q_t$). The first component of the right-hand side of this formula can be bounded via the following lemma:

**Lemma 4.** *Suppose that $\eta > 0$, $\ell \in \mathbb{R}^K$ and $w \in \mathcal{P}(K)$ satisfy $\eta |\ell(i)| \leq \frac{1-\alpha}{4} w(i)^{\alpha-1}$ for all $i \in [K]$. We then have $\langle \ell, w - w' \rangle - \frac{1}{\eta} D(w', w) \leq \frac{4\eta}{1-\alpha} \sum_{i \in [K]} w(i)^{2-\alpha} \ell(i)^2$.*

This follows, e.g., directly from the first part of Lemma 9 given by Ito et al. [2024]. To check the sufficient conditions for applying Lemma 4, we use the following lemma:

**Lemma 5.** *It holds for any $x \in \mathcal{X}$ and $i \in [K]$ that*

$$\phi(x, i)^\top V(p_t)^{-1} \phi(x, i) \leq \min \left\{ \frac{\lambda_0}{\gamma}, \frac{1}{(1-\gamma)g(x)q_t(x, i)} \right\}. \tag{6}$$

*Consequently, we have*

$$\left| \hat{\ell}_t(x, i) \right| = \left| \left\langle \hat{\theta}_t, \phi(x, i) \right\rangle \right| \leq \sqrt{\frac{\lambda_0}{\gamma}, \min \left\{ \frac{\lambda_0}{\gamma}, \frac{1}{(1-\gamma)g(x)q_t(x, i)} \right\}}. \tag{7}$$

Combining Lemmas 4 and 5, we obtain the following:

**Lemma 6.** *Suppose that $\alpha \geq 1/2$ and that $\gamma$ is given as (5). Then it holds for any $t$ and $x \in \mathcal{X}$ that*

$$\left\langle \hat{\ell}_t(x), q_t(x) - q_{t+1}(x) \right\rangle - \frac{1}{\eta(x)} D(p_{t+1}(x), p_t(x)) \leq \frac{4\eta(x)}{\beta} \sum_{i \in [K]} q_t(x, i)^{2-\alpha} \hat{\ell}_t(x, i)^2. \tag{8}$$

Further, the expectation of the right-hand side of (8) can be bounded as in the following, which is the key lemma for leading to an improved regret bound of $O(\sqrt{dT \log_+ \frac{KS}{d}})$.

**Lemma 7.** *We have*

$$\mathbf{E} \left[ \eta(X_0) \sum_{i \in [K]} q_t(X_0, i)^{2-\alpha} \hat{\ell}_t(X_0, i)^2 \right] \leq \frac{1}{1-\gamma} \min \left\{ d \cdot \sup_{x \in \mathcal{X}} \eta(x), \ d^\alpha \cdot \sup_{x \in \mathcal{X}} \frac{\eta(x)}{g(x)^\beta} \right\}. \tag{9}$$

*Proof.* From the definition of $\hat{\ell}_t(x, i)$, for any fixed $x \in \mathcal{X}$ and $i \in [K]$, we have

$$\mathbf{E} \left[ \hat{\ell}_t(x, i)^2 \right] = \mathbf{E} \left[ \left( \ell_t \phi(X_t, I_t)^\top V(p_t)^{-1} \phi(x, i) \right)^2 \right] \leq \mathbf{E} \left[ \left( \phi(X_t, I_t)^\top V(p_t)^{-1} \phi(x, i) \right)^2 \right]$$

$$= \mathbf{E} \left[ \phi(x, i)^\top V(p_t)^{-1} \phi(X_t, I_t) \phi(X_t, I_t)^\top V(p_t)^{-1} \phi(x, i) \right]$$

$$= \phi(x, i)^\top V(p_t)^{-1} V(p_t) V(p_t)^{-1} \phi(x, i)$$

$$= \phi(x, i)^\top V(p_t)^{-1} \phi(x, i) \leq \frac{1}{1-\gamma} \phi(x, i)^\top V(q_t)^{-1} \phi(x, i). \tag{10}$$

Denote $\eta_1 = \sup_{x \in \mathcal{X}} \eta(x)/g(x)^\beta$. Then, from (10), we have

$$
\begin{aligned}
[\text{LHS of (9)}] &\leq \frac{\eta_1}{1-\gamma} \mathbf{E}\left[ g(X_0)^\beta \sum_{i \in [K]} q_t(X_0, i)^{1+\beta} \phi(X_0, i)^\top V(q_t)^{-1} \phi(X_0, i) \right] \\
&= \frac{\eta_1}{1-\gamma} \sum_{x \in \mathcal{X}} g(x)^{1+\beta} \sum_{i \in [K]} q_t(x, i)^{1+\beta} \phi(x, i)^\top V(q_t)^{-1} \phi(x, i) \\
&= \frac{\eta_1}{1-\gamma} \sum_{x \in \mathcal{X}} \sum_{i \in [K]} u(x, i)^\beta v(x, i), \quad\quad\quad\quad\quad (11)
\end{aligned}
$$

where we define $u(x, i) = g(x) q_t(x, i)$ and $v(x, i) = g(x) q_t(x, i) \phi(x, i)^\top V(q_t)^{-1} \phi(x, i)$. We then have (i) $u(x, i) \geq 0$ for all $x \in \mathcal{X}$ and $i \in [K]$; $\sum_{x' \in \mathcal{X}} \sum_{i' \in [K]} u(x', i') = 1$, (ii) $\sum_{x' \in \mathcal{X}} \sum_{i' \in [K]} v(x', i') = d$, and (iii) $0 \leq v(x, i) \leq 1$ for all $x \in \mathcal{X}$ and $i \in [K]$. Indeed, the condition (i) is clear from the fact that $u(x, i)$ is a probability mass function over $\mathcal{X} \times [K]$. The condition (ii) follows from

$$
\begin{aligned}
\sum_{x' \in \mathcal{X}} \sum_{i' \in [K]} v(x', i') &= \text{tr}\left( V(q_t)^{-1} \sum_{x \in \mathcal{X}} \sum_{i \in [K]} g(x) q_t(x, i) \phi(x, i) \phi(x, i)^\top \right) \\
&= \text{tr}\left( V(q_t)^{-1} V(q_t) \right) = \text{tr}(I_d) = d.
\end{aligned}
$$

The condition (iii) follows from Lemma 8 and the fact that as $g(x) q_t(x, i) \phi(x, i) \phi(x, a)^\top \succeq V(q_t)$. Let $U \subseteq \mathcal{X} \times [K]$ be the top-$d$ subset with respect to the values of $u(x, a)$, i.e., let $U$ be such that $|U| = d$ and $u(x, i) \geq u(x', i')$ for any $(x, i) \in U$ and any $(x', i') \in (\mathcal{X} \times [K]) \setminus U$. Then, when we consider maximizing $\sum_{x \in \mathcal{X}} \sum_{i \in [K]} u(x, i)^\beta v(x, i)$ subject to the constraint of (ii) and (iii) on $v$, the maximum is attained by $v(x, i) = \mathbf{1}[(x, i) \in U]$. We hence have $\sum_{x \in \mathcal{X}} \sum_{i \in [K]} u(x, i)^\beta v(x, i) \leq \sum_{(x,i) \in U} u(x, i)^\beta \leq |U|^{1-\beta} \left( \sum_{(x,i) \in U} u(x, i) \right)^\beta \leq d^\alpha$, where the second inequality follows from Hölder's inequality and the last inequality follows from the condition (i). By combining this with (11), we obtain $[\text{LHS of (9)}] \leq \frac{\eta_1 d^\alpha}{1-\gamma}$. Similarly, denoting $\eta_0 = \sup_{x \in \mathcal{X}} \eta(x)$, we obtain $[\text{LHS of (9)}] \leq \frac{\eta_0}{1-\gamma} \mathbf{E}\left[ \sum_{i \in [K]} q_t(X_0, i) \phi(X_0, i)^\top V(q_t)^{-1} \phi(X_0, i) \right] = \frac{\eta_0}{1-\gamma} \mathbf{E}\left[ \text{tr}\left( V(q_t)^{-1} \sum_{i \in [K]} q_t(X_0, i) \phi(X_0, i) \phi(X_0, i)^\top \right) \right] = \frac{\eta_0}{1-\gamma} \sum_{x \in \mathcal{X}} \sum_{i \in [K]} v(x, i) = \frac{\eta_0 d}{1-\gamma}$, which completes the proof. $\quad\square$

Now we are ready to provide an upper bound on regret.

We can easily see that Theorem 3 is a direct consequence of Lemmas 3, 6 and 7.

*Proof of Corollary 1.* Suppose $\eta(x) = \eta' g(x)^\beta$ with $\eta' = \Theta\left( \sqrt{\frac{\beta (KS)^\beta}{\alpha d^\alpha T}} \right)$ and $\beta = \Theta\left( 1/\left( \log_+ \left( \frac{KS}{d} \right) \right) \right)$. We then have $R_T = O\left( \frac{\eta' d^\alpha T}{\beta} + \gamma T + \frac{K^\beta}{\eta'^\alpha} \sum_{x \in \mathcal{X}} g(x)^\alpha \right) = O\left( \frac{\eta' d^\alpha T}{\beta} + \gamma T + \frac{(KS)^\beta}{\eta'^\alpha} \right) = O\left( \sqrt{\frac{dT}{\alpha\beta} \left( \frac{KS}{d} \right)^\beta} + \lambda_0 (\eta')^{1/\alpha} T \right) = O\left( \sqrt{dT \log_+ \left( \frac{KS}{d} \right)} + \lambda_0 \sqrt{T^{1-\beta}} \right)$.

Suppose $\eta(x) = \eta = \Theta\left( \sqrt{\frac{\beta K^\beta}{\alpha d T}} \right)$ for all $x \in \mathcal{X}$ and $\beta = \Theta\left( \frac{1}{\log K} \right)$. We then have $R_T = O\left( \frac{\eta d T}{\beta} + \gamma T + \frac{K^\beta}{\eta^\alpha} \right) = O\left( \sqrt{\frac{d K^\beta T}{\alpha\beta}} + \lambda_0 \min\left\{ \sqrt{\frac{\beta K^\beta T}{d}}, L^{\frac{1-\alpha}{\alpha}} \left( \frac{\beta K^\beta}{d} \right)^{\frac{1}{2\alpha}} T^{1-\frac{1}{2\alpha}} \right\} \right) = O\left( \sqrt{dT \log K} + \lambda_0 \min\left\{ \sqrt{\frac{T}{d \log K}}, L^{\frac{1-\alpha}{\alpha}} \left( \frac{1}{d \log K} \right)^{\frac{1}{2\alpha}} \sqrt{T^{1-\beta}} \right\} \right) = O\left( \sqrt{dT \log K} + \lambda_0 \min\left\{ \sqrt{\frac{T}{d \log K}}, L^{\frac{\beta}{\alpha}} \sqrt{T^{1-\beta}} \right\} \right). \quad\square$

### 3.4 Regret lower bound

The following theorem implies that the regret upper bound given in Corollary 1 achieved by the Algorithm 1 is tight for $S \leq K \leq 2^d$.

**Theorem 4.** *Suppose any $d' \geq 1$, $S \geq 1$ and $T = \Omega(d'^2 S)$. Then, for any algorithm for contextual linear bandit problems with $K = 2^{d'}$, $d = d'S$, and $|\mathcal{X}| = S$ there exists a problem instance for which $R_T = \Omega\left(d'\sqrt{ST}\right)$.*

This theorem implies a regret lower bound of $\Omega\left(\sqrt{dT \log_+ \left(K \min\left\{1, \frac{S}{d}\right\}\right)}\right)$. To see this, we first note Theorem 4 implies that, if some $d'$ satisfies $K \geq 2^{d'}$ and $d \geq d'S$, we can obtain a regret lower bound of $R_T = \Omega(d'\sqrt{ST})$. Let $(d, K, S)$ be an arbitrary given parameter set that satisfies $K \leq 2^d \leq K^S$. We then have $\log_2 K \leq d \leq S \log_2 K$. Define $d' := \lfloor \log_2 K \rfloor$ and $S' := \lfloor d/\log_2 K \rfloor \leq S$. Then, as we have $K \geq 2^{d'}$ and $d \geq S' \log_2 K \geq S'd' = \Omega(d)$, from Theorem 4, we obtain a regret lower bound of $R_T = \Omega(d'\sqrt{S'T}) = \Omega(\sqrt{S'd'Td'}) = \Omega(\sqrt{dTd'})$. By combining this with $d' = \Omega(\log K) = \Omega\left(\log_+\left(K\min\{1, \frac{S}{d}\}\right)\right)$, we obtain $R_T = \Omega\left(\sqrt{dT \log_+\left(K\min\{1, \frac{S}{d}\}\right)}\right)$. We hence have the following lower bound:

**Corollary 2.** *For any $(d, K, S)$ such that $K \leq 2^d \leq K^S$ and for any algorithm for contextual linear bandit problems, there exists a problem instance for which $R_T = \Omega\left(\sqrt{dT \log_+\left(K \min\left\{1, \frac{S}{d}\right\}\right)}\right)$.*

*Proof of Theorem 4.* Theorem 4 can be shown by using the result of Dani et al. [2008, Theorem 3]. They provide a lower bound of $\Omega(d\sqrt{T})$ for (non-contextual) linear bandit problems with $K = 2^d$. We use $S$ copies of their problem instance of the dimensionality $d'$, to prove Theorem 4. The context $X_t$ is drawn from uniform distribution over $\mathcal{X} = [S]$, each element of which corresponds to one of the copies of the linear bandit instance. Then, for any $x \in \mathcal{X}$, the number of rounds $t \leq T$ at which $X_t = x$ is of $\Omega(T/S)$ with a probability at least $1/2$. Hence, the expected cumulative regret suffered for rounds at which $X_t = x$ is of $\Omega(d'\sqrt{T/S})$ for each $x \in \mathcal{X}$. By summing this for all $x \in \mathcal{X}$, we obtain the lower bound of $\Omega(Sd'\sqrt{T/S}) = \Omega(d'\sqrt{ST})$. Note that features mapping $\phi$ need to be designed so that $\phi(x, i)$ and $\phi(x', i')$ are orthogonal for any $x \neq x'$ and $i, i' \in [K]$. We can satisfy this condition by setting the dimension of the entire feature space to $d = d'S$. $\qquad\square$

## 4 Conclusion

In this study, we investigated the minimax regret in the contexts of the multi-armed bandit with expert advice and contextual linear bandit problems. For the former, we established a regret lower bound of $O(\sqrt{KT \log \frac{N}{K}})$ in the setting where the player selects an expert before observing expert advice. This bound matches, up to a constant factor, the upper bound provided by Kale [2014]. Additionally, for the contextual linear bandit problem, we proposed an algorithm that achieves a regret upper bound of $O(\sqrt{dT \log(K \min\{1, \frac{S}{d}\})})$. As illustrated in Table 1, this upper bound aligns with the lower bound under certain conditions on the parameters $(K, S, D)$.

Remaining challenges in the problem with expert advice include establishing similar lower bounds when the player can observe expert advice before decision-making in each round. Furthermore, determining the minimax regret in a broader parameter setting for (contextual) linear bandits remains an open problem. Relaxing Assumption 1 on the contextual distribution is also an important direction for enhancing practical applicability.

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

## A  Auxiliary lemma

**Lemma 8.** *For any positive-definite matrix $A \in \mathbb{R}^{d \times d}$ and a real vector $x \in \mathbb{R}^d$, if $A \succeq xx^\top$, it holds that $xA^{-1}x \leq 1$.*

*Proof.* From the assumption of $A \succeq xx^\top$, we have

$$x^\top A^{-1} x = (A^{-1}x)^\top A (A^{-1}x) \geq (A^{-1}x)^\top xx^\top (A^{-1}x) = (x^\top A^{-1}x)^2,$$

which implies $x^\top A^{-1}x \leq 1$. □

## B  Omitted lemmas and proofs in Section 2

**Lemma 9.** *Suppose $m = 1$. It holds for any $N', T^* \in \mathbb{N}$ and $\varepsilon \in [0, 1/2]$ that*

$$D_{\mathrm{KL}}\left(\mathcal{D}_{T^*}(\varepsilon, 1) \,\|\, \mathcal{D}_{T^*}(\varepsilon)\right) \leq \ln\left(\frac{N' - 1 + \left(1 + 4\varepsilon^2\right)^{T^*}}{N'}\right) \leq \frac{\left(1 + 4\varepsilon^2\right)^{T^*} - 1}{N'}.$$

*Consequently, if $T^* \leq \frac{\ln(N'/4)}{4\varepsilon^2}$, then the above value of the KL divergence is at most $1/4$.*

*Proof.* For $(e, \ell)$, let $b_1 \in \{0, 1\}$, $(c_v)_{v \in [N']} \in \{0, 1\}^{N'}$ be binary variables such that $\ell((1, 0)) = b_u$, $\ell((1, 1)) = 1 - b_u$, and $e((1, v)) = |b_u - c_v|$. Then, all the values of $(e, \ell)$ are determined by $\ell(0), b_1$ and $(c_v)_{v \in [N']}$. When $(e, \ell)$ follows $\mathcal{D}(\varepsilon)$, then $(\ell(0), b_1, (c_v)_{v \in [N']})$ follows $\mathcal{E}_0 := \mathrm{Ber}((1 - \varepsilon)/2) \times \mathrm{Ber}(1/2) \times (\mathrm{Ber}(1/2))^{N'}$. On the other hand, when $(e, \ell)$ follows $\mathcal{D}(\varepsilon, 1, v^*)$ then $(\ell(0), b_1, (c_v)_{v \in [N']})$ follows $\mathcal{E}_{v^*} := \mathrm{Ber}((1 - \varepsilon)/2) \times \mathrm{Ber}(1/2) \times \mathcal{F}_{N'}(\varepsilon, v^*)$, where we define $\mathcal{F}_{v^*} = (\mathrm{Ber}(1/2))^{v^* - 1} \times \mathrm{Ber}(1/2 + \varepsilon) \times (\mathrm{Ber}(1/2))^{N' - v^*}$. That is, if $(c_v)_{v \in [N']}$ follows $\mathcal{F}_{v^*}$, the $v^*$-th element $c_{v^*}$ follows $\mathrm{Ber}(1/2)$ and the other elements follow $\mathrm{Ber}(1/2)$ independently. Hence, from the data processing inequality, recalling the definition of $\mathcal{D}_{T^*}(\varepsilon)$ and $\mathcal{D}_{T^*}(\varepsilon, v^*)$ given in Section 2.3, we obtain

$$D_{\mathrm{KL}}\left(\mathcal{D}_{T^*}(\varepsilon, 1) \,\|\, \mathcal{D}_{T^*}(\varepsilon)\right) \leq D_{\mathrm{KL}}\left(\frac{1}{N'}\sum_{v^* \in N'}(\mathcal{E}_{v^*})^{T^*} \,\|\, (\mathcal{E}_0)^{T^*}\right)$$

$$= D_{\mathrm{KL}}\left(\frac{1}{N'}\sum_{v^* \in N'}(\mathcal{F}_{v^*})^{T^*} \,\|\, \left((\mathrm{Ber}(1/2))^{N'}\right)^{T^*}\right). \tag{12}$$

Let $p_0 : \{0, 1\}^{T^* \times N'} \to \mathbb{R}$ and $p_v : \{0, 1\}^{T^* \times N'}$ be the probability mass functions for $\left((\mathrm{Ber}(1/2))^{N'}\right)^{T^*}$ and $(\mathcal{F}_{v^*})^{T^*}$. Then, from the definition of the KL divergence, we have

$$[\text{RHS of (12)}] = \frac{1}{N'}\sum_{v^* \in [N']} \mathop{\mathbb{E}}_{c \sim (\mathcal{F}_{v^*})^{T^*}}\left[\ln\left(\frac{1}{N'}\sum_{v \in [N']}\frac{p_v(c)}{p_0(c)}\right)\right]$$

$$\leq \frac{1}{N'}\sum_{v^* \in [N']}\ln\left(\frac{1}{N'}\sum_{v \in [N']}\mathop{\mathbb{E}}_{c \sim (\mathcal{F}_{v^*})^{T^*}}\left[\frac{p_v(c)}{p_0(c)}\right]\right), \tag{13}$$

where we used Jensen's inequality and the fact that $\ln(x)$ is a concave function. The ratio $\frac{p_{v^*}(c)}{p_0(c)}$ of probabilities can be expressed as

$$\frac{p_v(c)}{p_0(c)} = \prod_{t \in [T^*]}(\mathbf{1}[c_{tv} = 0] \cdot (1 - 2\varepsilon) + \mathbf{1}[c_{tv} = 1] \cdot (1 + 2\varepsilon)) = \prod_{t \in [T^*]}(1 + (4c_{tv} - 2)\varepsilon).$$

Hence, if $v \neq v^*$ then

$$\mathop{\mathbb{E}}_{c \sim (\mathcal{F}_{v^*})^{T^*}}\left[\frac{p_v(c)}{p_0(c)}\right] = \prod_{t \in [T^*]}\mathop{\mathbb{E}}_{(c_{tv}) \sim \mathrm{Ber}(1/2)}(1 + (4c_{tv} - 2)\varepsilon) = 1,$$

where we used the condition that elements of $(c_{tv})$ are independent. Further, if $v = v^*$, we then have

$$\mathbf{E}_{c \sim (\mathcal{F}_{v^*})^{T^*}} \left[ \frac{p_v(c)}{p_0(c)} \right] = \prod_{t \in [T^*]} \mathbf{E}_{(c_{tv}) \sim \mathrm{Ber}(1/2+\varepsilon)} (1 + (4c_{tv} - 2)\varepsilon) = \left(1 + 4\varepsilon^2\right)^T.$$

Therefore, we have

$$[\text{RHS of (13)}] = \frac{1}{N'} \sum_{v^* \in [N']} \ln \left( \frac{N' - 1 + \left(1 + 4\varepsilon^2\right)^{T^*}}{N'} \right) = \ln \left( \frac{N' - 1 + \left(1 + 4\varepsilon^2\right)^{T^*}}{N'} \right),$$

which completes the proof. $\qquad\square$

## B.1 Proof of Theorem 1

*Proof.* For each $u \in [m]$, let $\mathcal{T}_u(\mathcal{A}, \mathcal{D}(\varepsilon))$ denote the expected value of the number of rounds in which the algorithm $\mathcal{A}$ observes $\ell_t((u, 0))$ or $\ell_t((u, 1))$ before termination. Then, we will show that $\mathcal{T}_u(\mathcal{A}, \mathcal{D}(\varepsilon)) \geq T^*/2$ under the assumption that $\mathcal{A}$ is an $(\varepsilon/2, 0.05)$-PAC algorithm.

Fix any $u \in [m]$ and suppose that $\mathcal{A}$ is an $(\varepsilon/2, 0.05)$-PAC BEI algorithm for problems with general $m$. Then we can construct an $(\varepsilon/2, 0.05)$-PAC BEI algorithm $\mathcal{A}'$ for problems with $m = 1$ such that $\mathcal{T}(\mathcal{A}', \mathcal{D}'(\varepsilon)) \leq \mathcal{T}_u(\mathcal{A}, \mathcal{D}(\varepsilon))$, where $\mathcal{D}'(\varepsilon)$ and $\mathcal{D}(\varepsilon)$ represent distributions with $(m = 1, N' = N')$ and $(m = m, N' = N')$, respectively. To show this, consider the following procedure for solving BEI for $m = 1$ based on $\mathcal{A}$: Run algorithm $\mathcal{A}$. When algorithm $\mathcal{A}$ pulls an arm $a_t$ other than $(u, 0)$ or $(u, 1)$, then generate $e_t$ from $\mathcal{D}(\varepsilon)$ and $\ell(a_t)$ from $\mathrm{Ber}(1/2)$ independently of $e_t$, and feed them to $\mathcal{A}$. When algorithm $\mathcal{A}$ pulls $(u, 0)$ or $(u, 1)$, then query the instance of $m = 1$ and feed the observed losses to $\mathcal{A}$. Then, this construction provide an $(\varepsilon/2, 0.05)$-PAC BEI algorithm for the instance of $m = 1$, and if the instance is associated with $\mathcal{D}'(\varepsilon)$, the number of queries to this distributions has the expected value of $\mathcal{T}_u(\mathcal{A}, \mathcal{D}(\varepsilon))$. We hence have $\mathcal{T}(\mathcal{A}', \mathcal{D}'(\varepsilon)) \leq \mathcal{T}_u(\mathcal{A}, \mathcal{D}(\varepsilon))$.

From Lemma 2, we have $\mathcal{T}(\mathcal{A}', \mathcal{D}'(\varepsilon)) \geq T^*/2$. As we have $\mathcal{T}(\mathcal{A}, \mathcal{D}(\varepsilon)) \geq \sum_{u \in [m]} \mathcal{T}_u(\mathcal{A}, \mathcal{D}(\varepsilon)) \geq \sum_{u \in [m]} \mathcal{T}(\mathcal{A}', \mathcal{D}'(\varepsilon))$, we obtain $\mathcal{T}(\mathcal{A}, \mathcal{D}(\varepsilon)) \geq mT^*/2$. $\qquad\square$

# C Omitted proofs in Section 3

## C.1 Proof of Lemma 3

*Proof.* Fix $\pi^* : \mathcal{X} \to [K]$. From the definitions of $p_t$ in (2) and $X_0$, and the fact that $\hat{\theta}_t$ is an unbiased estimator of $\theta_t$ as shown in (4), we have

$$
\begin{aligned}
R_T(\pi^*) &= \mathbf{E}\left[ \sum_{t=1}^{T} \langle \theta_t, \phi(X_t, I_t) - \phi(X_t, \pi^*(X_t)) \rangle \right] \\
&= \mathbf{E}\left[ \sum_{t=1}^{T} \left\langle \theta_t, \sum_{i \in [K]} p_t(X_t, i)\phi(X_t, i) - \phi(X_t, \pi^*(X_t)) \right\rangle \right] \\
&\leq \mathbf{E}\left[ \sum_{t=1}^{T} \left\langle \theta_t, \sum_{i \in [K]} q_t(X_t, i)\phi(X_t, i) - \phi(X_t, \pi^*(X_t)) \right\rangle \right] + 2\gamma T \\
&= \mathbf{E}\left[ \sum_{t=1}^{T} \left\langle \theta_t, \sum_{i \in [K]} q_t(X_0, i)\phi(X_0, i) - \phi(X_0, \pi^*(X_0)) \right\rangle \right] + 2\gamma T \\
&= \mathbf{E}\left[ \sum_{t=1}^{T} \left\langle \hat{\theta}_t, \sum_{i \in [K]} q_t(X_0, i)\phi(X_0, i) - \phi(X_0, \pi^*(X_0)) \right\rangle \right] + 2\gamma T \\
&= \mathbf{E}\left[ \sum_{t=1}^{T} \left( \left\langle \hat{\ell}_t, q_t(X_0) \right\rangle - \hat{\ell}_t(\pi^*(X_0)) \right) \right] + 2\gamma T.
\end{aligned}
\tag{14}
$$

From [Lattimore and Szepesvári, 2020, Theorem 28.5], for any $x \in \mathcal{X}$, $(q_t(x))_{t \in [T]}$ defined by the FTRL procedure (2) satisfies

$$\sum_{t=1}^{T} \left( \left\langle \hat{\ell}_t, q_t(X_0) \right\rangle - \hat{\ell}_t(\pi^*(X_0)) \right)$$

$$\leq \sum_{t=1}^{T} \left( \left\langle \hat{\ell}_t(x), q_t(x) - q_{t+1}(x) \right\rangle - \frac{1}{\eta(x)} D(p_{t+1}(x), p_t(x)) \right) - \frac{\psi(q_1(x))}{\eta(x)}. \tag{15}$$

Combining (14) and (8) with the fact that $\psi(q_1(x)) = \psi(\mathbf{1}/K) = -\frac{K^\beta - 1}{\alpha}$, we obtain the desired inequality. $\qquad \square$

## C.2 Proof of Lemma 5

*Proof.* From the definition of $p_t$ in (2), as we have $V(p_t) \succeq \gamma V(p_0)$, we have

$$\phi(x,i)^\top V(p_t)^{-1} \phi(x,i) \leq \phi(x,i)^\top (\gamma V(p_0))^{-1} \phi(x,i) = \frac{1}{\gamma} \phi(x,i)^\top V(p_0)^{-1} \phi(x,i) \leq \frac{\lambda(p_0)}{\gamma} = \frac{\lambda_0}{\gamma}. \tag{16}$$

Further, as we have $V(p_t) \succeq (1-\gamma)V(q_t) \succeq (1-\gamma)g(x)q_t(x,i)\phi(x,i)\phi(x,i)^\top$, from Lemma 8, we have $\phi(x,i)^\top V(p_t)^{-1} \phi(x,i) \leq \frac{1}{(1-\gamma)g(x)q_t(x,i)}$. This, together with (16), yields (6). From the definition (3) of $\hat{\theta}_t$, we have

$$\left| \hat{\ell}_t(x,i) \right| = \left| \left\langle \hat{\theta}_t, \phi(x,i) \right\rangle \right| = \left| \ell_t \phi(X_t, I_t)^\top V(p_t) \phi(x,i) \right|$$

$$\leq \sqrt{\phi(X_t, I_t)^\top V(p_t) \phi(X_t, I_t) \phi(x,i)^\top V(p_t) \phi(x,i)},$$

where we used Cauchy-Schwarz inequality and the assumption that $|\ell_t| \leq 1$. From this and (6), we have (7). $\qquad \square$

