# OpenReview forum: "On the Minimax Regret for Contextual Linear Bandits and Multi-Armed Bandits with Expert Advice"
_NeurIPS.cc/2024/Conference — NeurIPS 2024 poster_

### Official Review · Reviewer_nf4R · 2024-07-07

**Soundness:** 3
**Presentation:** 3
**Contribution:** 3
**Rating:** 6
**Confidence:** 3

**Summary:**

This paper studies the problem of MAB with expert advice and the contextual linear MAB. Specificially, the authors proposed a lower bound of $\Omega(\sqrt{KT\log\frac{N}{K}})$, improving upon previously known lower bound $\Omega(\sqrt{KT\frac{\log N}{\log K}})$ and showing that the previous upper bound $O(\sqrt{KT\log\frac{N}{K}})$ is tight. For the second problem, the authors design an algorithm based on FTRL with Tsallis entropy regularizer and achieve $O(\sqrt{dT \log K(\min(1,S/d))})$ regret bound. A matching lower bound is also proven (under certain condition of $K$) based on an extension of the previous lower bound constructed for linear bandits.

**Strengths:**

- The paper bridges the gap between the lower and the upper bound for MAB with expert advice, solving an open problem in this field.
- While it is not technically hard to show that a regret minimization algorithm can be transformed to a best expert identification algorithm, the reduction with a specific construction of the expert problem instance leads to a matching lower bound, which is interesting to me.
- While both the algorithm and the lower bound construction for contextual linear case are kind of standard from a technical perspective, the matching regret bound is good to know.

**Weaknesses:**

While I do not have much concern for the MAB with expert advice problem, for the contextual linear bandit problem, one of my concern is that the algorithm only works for the finite context case, which is very restrictive in real applications. In addition, the implementation of the algorithm requires the knowledge of the context distribution, which is also a limitation of the algorithm. As for the regret bound, while the leading term matches the lower bound, the optimality of the additonal term is unclear. Specifically, in the second bound, $L$ may also be related to $T$ (e.g. $T^{-c}$), making the additional term dominant.

**Questions:**

- Whether Algorithm 1 can be extended to infinite / unknown context case?
- Whether the additional term in the upper bound for contextual linear bandits is unavoidable?

**Limitations:**

See "weakness" and "questions".

---

> ### Author Rebuttal · Authors · 2024-08-07
>
> > one of my concern is that the algorithm only works for the finite context case, which is very restrictive in real applications.
>
> The proposed algorithm (Algorithm 1) can also work for the infinite context case,
> in which it enjoys the second regret upper bound in Corollary 1 (line 222):
> $R_T = O \left( \sqrt{dT \log K} + \lambda_0 \sqrt{T / (d \log K)} \right)$.
> In fact,
> this regret upper bound does not include $S = |\mathcal{X}|$ or $L$,
> and
> the value of $g(x)$ is not required to define $\eta(x)$ in showing this second upper bound.
> Hence,
> we do not require Assumption 1 (line 192) to show this bound.
>
> In the infinite context case,
> however,
> further challenges regarding the computational complexity and practicality of the algorithm should be noted.
> For example,
> we need to compute $V(p_t)$ (defined in line 195) in the algorithm as it appears in the definition of $\hat{\theta}_t$ in (3),
> which tend to be computationally expensive,
> depending on the computational model and the setup of distributions.
> We also require Assumption 2 (line 196) as well.
>
>
> >  In addition, the implementation of the algorithm requires the knowledge of the context distribution, which is also a limitation of the algorithm.
>
> We agree with this comment.
> Relaxing this assumption is an important future research direction.
> We believe that the technique of *Matrix Geometric Resampling* [NO20] can be used to relax the assumptions that the context distribution is known,
> and to consider settings where any number of samples from the distribution can be accessed.
> However,
> as long as this is employed,
> an extra $\sqrt{\log T}$ factor seems inevitable.
> To further relax the assumption,
> [LWZ23] have proposed algorithms that do not even require access to a simulator from which the learner can draw samples from the context distribution.
> For such a setup,
> there is no known algorithm that achieve a $(\log K)$-dependent bound or that avoid extra $\sqrt{\log T}$ factors,
> to our knowledge.
>
>
>
> > As for the regret bound, while the leading term matches the lower bound, the optimality of the additonal term is unclear. Specifically, in the second bound, $L$ may also be related to $T$ (e.g. $T^{-c}$), making the additional term dominant.
>
> The parameter $L$ is defined on the basis of the context distribution,
> independent of $T$,
> in Assumption 1 and does not grow with $T$ after fixing the problem instance.
> Thus, after fixing the context distribution arbitrarily, the additional term will not be dominant in a regime in which $T$ is sufficiently large.
> However, conversely, if we consider the regime of considering the worst-case context distribution after fixing $T$,
> then the additional term can be dominant,
> as the reviewer suggested.
>
> We here note that the additional term in the second bound of Corollary 1
> is the **minimum** of $\lambda_0 \sqrt{\frac{T}{d \log K}}$ and $\lambda_0 L^{\beta/\alpha} \sqrt{T^{1-\beta}}$,
> and hence large $L$ is not necessarily problematic.
> However,
> the parameter $\lambda_0$,
> which is defined in Assumption 2 on the basis of the context distribution and exploration policy $p_0$,
> can be arbitrarily large depending on the context distribution,
> which might make the additional term dominant.
>
>
> > Whether Algorithm 1 can be extended to infinite / unknown context case?
>
> Please refer to the response above.
>
> > Whether the additional term in the upper bound for contextual linear bandits is unavoidable?
>
> This is an open question at this time to our knowledge.
> However, if the main term can be $O(\sqrt{d^2 T \log T})$ instead of $O(\sqrt{d T \log (K \min \\{ 1, S/d \\}}))$,
> then we can remove the terms regarding $\lambda$ and $L$,
> as shown in Theorem 4 of [LWZ23].
>
>
> Reference
> * [LWZ23]H. Liu, C.-Y. Wei, and J. Zimmert. Bypassing the simulator: Near-optimal adversarial linear contextual bandits. NeurIPS2023.
> * [NO20] G. Neu and J. Olkhovskaya. Efficient and robust algorithms for adversarial linear contextual bandits. COLT2020.

---

### Official Review · Reviewer_sKYc · 2024-07-12

**Soundness:** 3
**Presentation:** 2
**Contribution:** 3
**Rating:** 6
**Confidence:** 2

**Summary:**

This paper studies the problem of bandit learning with expert advice. The main contributions are two refined regret bounds: (1) A matching lower regret bound of $\Omega(\sqrt{KT\log(N/K)})$ for multi-armed bandit problem; (2) $\Theta(\sqrt{dT\log(K,\min\{1,S/d\})})$ regret bound for contextual linear bandits.

**Strengths:**

This authors prove matching regret bounds for bandit problem with expert advice, which is significant enough for an acceptance.

**Weaknesses:**

My concern is about the writing. After reading the paper, I think the high-level intuitions of the construction for lower bounds could be clarified in one or two pages.

I also have a question about the proof of Lemma 3. In equation (15), a lemma in the bandit algorithm book is introduced to derive regret bounds. As far as I can see, the lemma only works for fixed $X_0$. However, in the learning process $p_t$ might depends on the historical contexts $X_0,X_1,X_2,...,X_{t-1}$. I am confused about the usage of the lemma and hope for some explanations.

**Questions:**

Please find my question in the comments above.

**Limitations:**

Yes.

---

> ### Author Rebuttal · Authors · 2024-08-07
>
> > I also have a question about the proof of Lemma 3. In equation (15), a lemma in the bandit algorithm book is introduced to derive regret bounds. As far as I can see, the lemma only works for fixed $X_0$. However, in the learning process $p_t$ might depends on the historical contexts $X_0, X_1, X_2, \ldots, X_{t-1}$. I am confused about the usage of the lemma and hope for some explanations.
>
> As mentioned in lines 227--229,
> the variable $X_0$ is a kind of *dummy* random variable that does not appear in the decision-making process or algorithms, but appear *only in the analysis*,
> and is defined to be independent of $X_1, X_2, \ldots, X_{T}$
> (and therefore is independent of any other variables including $p_t$).
> The approach of such an analysis is proposed by [NO20],
> in which this is referred to as the *ghost sample*.
> The idea and usage of this technique are explained around (1) of Section 2 of [NO20].
> Since $X_0$ is independent of all other random variables, it justifies the approach of marginalizing w.r.t. $X_0$ after evaluating for a fixed $X_0$.
> The revised version explains this more explicitly.
>
> Reference
> * [NO20] G. Neu and J. Olkhovskaya. Efficient and robust algorithms for adversarial linear contextual bandits. COLT202.

---

> ### Comment · Reviewer_sKYc · 2024-08-09
>
> I am still confused about the proof so I hope for more detailed analysis. The linear contextual bandit problem seems too dirty so we can consider contextual bandit problem instead.  For example, we assume there are  $K$ arms in $X$. In each round. Let the context distribution $D$ be the uniform distribution over all subsets of $X$ with size $K/2$. Consequently, with high probability $X_i\neq X_j$ for any $i\neq j$. We then choose the reward function $r_t(x)$ as a Bernoulli variable with mean $1/2$.
> Then the best response $\pi^*$ is the policy such that $\pi^*(X_t): = \arg\max_{x\in X_{t}}r_{t}(x)$. With hight probability, we have $\sum_{t=1}^T \max_{x\in X_{t}}r_t(x)=T$. However, to reach a sublinear regret, one needs to identify the arms with  reward $1$ in all but $o(T)$ rounds without any prior information. This seems impossible.
>
> Could you please explain more about this?

---

> > ### Author Response · Authors · 2024-08-10
> >
> > Thank you for your time and the opportunity to engage in further discussion.
> >
> > The construction you proposed would not suffice as a proof of the lower bound on regret.
> > As can be seen in Line 189,
> > in the definition of regret,
> > the operator of $\sup_{\pi^*}$ is placed **outside** of the expectation operator $\mathbb{E}[\cdot]$.
> > This means that
> > we take the maximum with respect to $\pi^*$ **after** taking the expectation with respect to the context.
> > Consequently,
> > the "optimal policy"
> > $\pi^*: \mathcal{ X } \rightarrow [K]$ cannot depend on the realization of contexts
> > but is the policy that maximize the expected total rewards.
> >
> > In the case of the problem instance you presented,
> > the policy $\pi^*$ is chosen to ensure that
> > $\pi^*(X_t) \in \arg\max_{x \in X\_t} r\_t(x)$
> > for (almost) all $t \in [T]$.
> > Such a policy $\pi^*: \mathcal{ X } \rightarrow [K]$ necessarily depends on the sequence $\\{ X\_t \\}_{t \in [T]}$.
> > Indeed,
> > for most randomly generated sequences $\\{ r\_t \\}\_{t \in [T]}$,
> > there exists no policy $\pi^*$ such that
> > $\pi^*(X_t) \in \arg\max\_{x \in X\_t} r\_t(x)$ $(t \in [T])$
> > for (almost) every possible realization of contexts $\\{ X\_t \\}\_{t \in [T]}$.
> >
> > In the literature on adversarial linear contextual bandits,
> > it appears standard to define regret by taking the maximum with respect to the comparator policy after taking the expectation,
> > as we have done in our paper.
> > For reference,
> > please see Section 2 of [LWZ23] and Section 2 of [NO20].
> > This can be interpreted as a comparison with the optimal policy chosen without knowledge of the realization of randomly generated contexts.
> >
> >
> >
> > Reference
> > * [LWZ23] H. Liu, C.-Y. Wei, and J. Zimmert. Bypassing the simulator: Near-optimal adversarial linear contextual bandits. NeurIPS2023.
> > * [NO20] G. Neu and J. Olkhovskaya. Efficient and robust algorithms for adversarial linear contextual bandits. COLT2020.

---

> > > ### Comment · Reviewer_sKYc · 2024-08-10
> > >
> > > Thanks for the response. I think there is some misunderstandings. $\pi^*$ is a mapping from the context to the arm, i.e., a mapping from $2^{[K]}$ to $[K]$, so why can't $\pi^*$ depend on the context?

---

> > > > ### Comment · Reviewer_sKYc · 2024-08-10
> > > >
> > > > Also another more concrete question: what is the optimal policy for the contextual bandit problem above?

---

> > > > > ### Author Response · Authors · 2024-08-10
> > > > >
> > > > > Thanks for the reply, and sorry for the confusion.
> > > > > We understand that $\pi^*$ is a mapping from $2^{[K]}$ to $[K]$,
> > > > > so of course,
> > > > > the **value** of $\pi^*(X\_t)$ depends on $X\_t$.
> > > > > My point is
> > > > > whether the **mapping** $\pi^*$ **itself** can change according to the realization of $\\{ X\_t \\}_{t \in [T]}$.
> > > > >
> > > > >
> > > > >
> > > > > As we mentioned,
> > > > > in the definition of regret,
> > > > > $\max_{\pi^*}$ is placed outside of $\mathbb{E}[ \cdot ]$,
> > > > > where the expectation is taken over all randomness.
> > > > > Thus,
> > > > > when considering bounds on regret,
> > > > > we focus on the value of
> > > > > $$
> > > > > \max_{\pi^*:2^{[K]} \rightarrow [K]} \mathbb{E}\_{\\{ X\_t \\}}\left[\sum_{t=1}^T r\_t( \pi^*(X\_t))\right],
> > > > > $$
> > > > > where the expectation is taken with respect to the contexts
> > > > > $\\{ X\_t \\}\_{t \in [T]} \sim {D}^T$.
> > > > > The "optimal policy" here is the policy that attain the maximum above.
> > > > > In contrast,
> > > > > we guess that you are focusing on the following value:
> > > > > $$
> > > > > \mathbb{E}\_{\\{ X\_t \\}}\left[ \max_{\pi^*:2^{[K]} \rightarrow [K]} \sum_{t=1}^T r\_t( \pi^*(X\_t))\right].
> > > > > $$
> > > > > In the latter,
> > > > > one is allowed to choose a different mapping $\pi^*$ depending on
> > > > > $\\{ X\_t \\}\_{t \in [T]}$,
> > > > > but not in the former.
> > > > > We consider this is an important difference.
> > > > > In the literature of adversarial linear contextual bandits,
> > > > > regret is defined with the former concept.
> > > > >
> > > > >
> > > > > > Also another more concrete question: what is the optimal policy for the contextual bandit problem above?
> > > > >
> > > > > The optimal policy is one that maximizes
> > > > > $\mathbb{E}\_{\\{ X\_t \\}}\left[\sum_{t=1}^T r\_t( \pi^*(X\_t))\right]$.
> > > > > As a simple example,
> > > > > let us consider the case of $T = 2$,
> > > > > $K = 4$,
> > > > > $X = \\{ 1, 2, 3, 4 \\}$,
> > > > > $r_1 = [1,1,-1,-1]$
> > > > > and
> > > > > $r_2 = [-1,-1,1,1]$.
> > > > > Suppose that
> > > > > $X_t$ is chosen from a uniform distribution over all subset of $X$ with size $2$.
> > > > > Then we can see that
> > > > > $$
> > > > > \mathbb{E}\_{\\{ X\_t \\}}\left[r\_1( \pi^*(X\_1)) + r\_2( \pi^*( X\_2 ))\right] = 0
> > > > > $$
> > > > > for any $\pi^*:2^[K] \rightarrow [K]$,
> > > > > which means that every $\pi^*$ is an optimal policy.
> > > > > In fact,
> > > > > as $X\_1$ and $X\_2$ follow an identical distribution,
> > > > > we have
> > > > > $$
> > > > > \mathbb{E}\_{\\{ X\_t \\}}\left[r\_1( \pi^*(X\_1)) + r\_2( \pi^*( X\_2 ))\right] = \mathbb{E}\_{\\{ X\_t \\}}\left[r\_1( \pi^*(X\_1)) + r\_2( \pi^*( X\_1 ))\right].
> > > > > $$
> > > > > If $\pi^*(X\_1) \in \\{ 1 , 2 \\}$,
> > > > > we have
> > > > > $r\_1( \pi^*(X\_1)) + r\_2(\pi^*( X\_1 )) = 1 + (- 1) = 0$.
> > > > > Otherwise,
> > > > > i.e.,
> > > > > if $\pi^*(X\_1) \in \\{ 3 , 4 \\}$,
> > > > > we have
> > > > > $r\_1( \pi^*(X\_1)) + r\_2(\pi^*( X\_1 )) = (-1) + 1 = 0$.
> > > > > Hence,
> > > > > the expected cumulative reward is $0$ for any $\pi^*$.

---

> > > > > > ### Comment · Reviewer_sKYc · 2024-08-10
> > > > > >
> > > > > > Thanks for the explanation. It is much more clear.

---

### Official Review · Reviewer_PXkJ · 2024-07-16

**Soundness:** 3
**Presentation:** 3
**Contribution:** 4
**Rating:** 7
**Confidence:** 3

**Summary:**

In this work, the authors tackle the existing gap between upper and lower bounds in bandits with expert advice. An existing lower bound scaled as $\Omega(\sqrt{KT \frac{\log N}{\log K}}$, while the state of the art of only provided a $O(\sqrt{KT \log (N/K)}$ bound (Kale 2014)
The authors close this gap by proposing a novel lower bound that matches the result of Kale 2014 and improves upon the lower bound of Seldin and Lugosi (2016).

Then, the authors also consider the problems of linear bandits and contextual linear bandits. For linear bandits, the authors improve upon the state of the art and provide tighter upper bounds, which match existing lower bounds for various relations of the dimension $d$ and the number of arms $K$.
The study of contextual linear bandits in this setting is fairly recent and there, the authors propose novel algorithm that improves upon the state of the art and a novel lower bound that matches their upper bound for certain combinations of dimensions $d$ and number of arms $K$.

**Strengths:**

This paper provides several significant improvements for variations around the multi-armed bandits. Notably, several of their contributions are lower bounds, one of which closed an open problem that had been open for 8 years.
The strategy used to solve this problem is a reduction from the best expert identification problem to the bandits with expert advice setting, and this approach will likely lead to other improvements in different fields.

In the field of linear bandits, the approach using FTRL with the correct tuning of Tsallis entropy has proved crucial to solving many bandit problems, not just in the adversarial setting but in the more challenging best-of-both-worlds regime.

In all of these sections, the authors provide detailed proofs that appear correct.

**Weaknesses:**

If anything, this paper could benefit from some discussions about the use of FTRL with Tsallis-Inf with tuning of $\alpha \in (1/2, 1)$ which is used to solve many variations of the multi-armed bandits problem (for example but not limited to: bandits with feedback graphs (Zimmert and Lattimore (2019), Ito et al. (2022), Dann, Wei and Zimmert (2023) or Decoupling explorations and exploitation in MAB (Rouyer and Seldin (2020),  Jin, Liu and Luo (2023)).

Using this framework is very beneficial in best-of-both-worlds settings rather than in the purely adversarial or purely stochastic regimes. Have you considered generalizing your results to that setting?

**Questions:**

See weaknesses.

**Limitations:**

Purely theoretical work. The scope of the results is clearly indicated in the theorems.

---

> ### Author Rebuttal · Authors · 2024-08-07
>
> > Using this framework is very beneficial in best-of-both-worlds settings rather than in the purely adversarial or purely stochastic regimes. Have you considered generalizing your results to that setting?
>
> Thanks for your suggestion.
> We believe that our results can be extended to the best-of-both-worlds (BOBW) setting if some difficulties are overcome.
> Challenging factors are inferred from the fact that there are only limited number of results achieving BOBW with FTRL using Tsallis entropy with $\alpha \neq 1/2$.
> The few such examples are [ITH24], [JLL23] and [RS20],
> which seem to require careful adjustment of learning rates that may depend on the feedback and output distributions so far,
> as is given,
> e.g.,
> in Algorithm 1 of [JLL23].
> They also seem to require some sort of stability condition, as given in equation (25) of [ITH24] and in Section C.3 of [JLL23].
> If we can overcome these challenges, we believe our results can be extended to the BOBW setting.
>
> Reference:
> * [ITH24] Shinji Ito, Taira Tsuchiya, and Junya Honda. Adaptive learning rate for follow-the-regularized-leader: Competitive analysis and best-of-both-worlds. COLT2024.
> * [JLL23] Tiancheng Jin, Junyan Liu, and Haipeng Luo. Improved best-of-both-worlds guarantees for multiarmed bandits: FTRL with general regularizers and multiple optimal arms. NeurIPS2023.
> * [RS20] Chloé Rouyer and Yevgeny Seldin. Tsallis-INF for decoupled exploration and exploitation in multiarmed bandits. COLT2020.

---

### Official Review · Reviewer_FSFk · 2024-07-27

**Soundness:** 3
**Presentation:** 2
**Contribution:** 3
**Rating:** 6
**Confidence:** 3

**Summary:**

The paper investigates two significant extensions of multi-armed bandit problems: multi-armed bandits with expert advice (MwE) and contextual linear bandits (CLB). For MwE, the authors close the gap between previously known upper and lower bounds, establishing a matching lower bound of $\\Omega\\left(\\sqrt{KT \\log \\frac{N}{K}}\\right)$, where $K$, $N>K$, and $T$ denote the number of arms, experts, and rounds, respectively. This claim seemingly resolves an open question posed by Seldin and Lugosi (2016), where a $\\Omega\\left(\\sqrt{KT \\frac{\\log N}{\\log K}}\\right)$ regret lower bound was shown. For CLB instead, the authors introduce an algorithm that achieves an improved upper bound of $O\\left( \\sqrt{dT \\log\\left(K \\min\\{1, \\frac{S}{d}\\}\\right)\} \right)$, where $d$ is the dimensionality of feature vectors, and $S$ is the size of the context space. The authors also provide a matching lower bound, confirming the minimax regret is $\\Theta\\left( \\sqrt{dT \\log\\left(K \\min\\{1, \\frac{S}{d}\\}\\right)} \\right)$. The results are achieved using the follow-the-regularized-leader (FTRL) approach using the negative Tsallis entropy regularizer for an appropriate tuning of its parameter, and carefully designed context-dependent learning rates.

**Strengths:**

This work provides relevant contributions to the theoretical understanding of multi-armed bandit problems in terms of the minimax regret.
Even if the use of the follow-the-regularized-leader (FTRL) approach with negative Tsallis entropy regularization follow from a previous line of work of improved regret bounds in other bandit problems, as made explicit by the authors in the introduction, the regret analysis of the proposed algorithm for CLB requires novel ideas in tuning the learning rate and more care in leveraging the possibility to tune the parameter of the Tsallis entropy.

More interestingly, the authors manage to provide an improved lower bound for a harder version of bandits with experts advice that matches the regret of the algorithm proposed by Kale (2014).
They do so via a proof argument that adapts ideas from the previous work on bandits with feedback graphs by Chen et al. (2023) to MwE, requiring a careful construction of hard problem instances for the latter problem. This construction is nontrivial, interesting, and could potentially help in the design of hard problem instances for proving lower bounds of other related problems.
Furthermore, introducing logarithmic factors in regret lower bounds is typically challenging and doing so enables a more complete understanding of the problems.
Similarly, the lower bound for CLB requires a clever adaptation of previous results.

**Weaknesses:**

While the lower bound for the MwE problem improves on prior results, it requires to consider a **harder** setting than the standard one by restricting the learner.
Specifically, they restrict the learner to selecting only experts rather than possibly choosing actions directly, and the learner can observe the expert advice only after making a decision.
The presentation is also somewhat misleading, as the authors claim to resolve the open problem left by Seldin and Lugosi (2016) for the “classical” problem of multi-armed bandits with expert advice while it is not exactly the case.
To fully address that open question, one needs to consider learners that may select actions irrespectively of the expert advice, and that observe the expert advice before choosing an action.
Nonetheless, the authors themselves make a clear point (rf. Remark 1) that this is indeed a more challenging setting, and my concern mainly revolves about the claims in the abstract and introduction.
It should also be clarified that proving the same lower bound for the standard setting remains an open problem that could be interesting to investigate.

Some proofs in the paper feel more like sketches rather than complete proofs. While they provide a good overview of the approach and main ideas, they lack detailed steps and thorough explanations. This makes it challenging for readers to fully verify the results and understand all nuances of the arguments.
For instance, the proof of Corollary 1 would benefit from a more detailed explanation of the steps; additionally, an intuitive explanation of the choice of the context-dependent learning rate would help the reader while possibly helping in adapting a similar idea in other related problem settings.
Another example is provided by the proof of Theorem 4, which is missing a more formal and explicit computation of the lower bound.

Finally, the results in this current work still do not achieve minimax optimality for the non-contextual linear bandits problem. Indeed, the only known lower bounds that match the upper bound provided in this work hold for the cases $K=d$ and $K=2^d$, respectively. The problem of proving a lower bound for arbitrary values of $K$ is still open and stating this explicitly would make the exposition of the contribution of this paper more transparent and clearer for the reader. A similar question could be made for the contextual version of the problem since the lower bound proved in this paper requires $2^{d/S} \\le K \\le 2^d$.

**Questions:**

- Could you clarify the doubts that were raised in the weaknesses above?
- Could you expand a bit on how the argument at lines 278-279 allows us to keep the generality of the values of $S$, $K$, and $d$ as claimed?
- Do you think the ideas from the different lower bounds for linear bandits could be adapted and combined to prove a regret lower bound for arbitrary values of $K$? Or is there a much larger technical hurdle that needs to be overcome?

Minor comments/typos:
- Line 99: The argmin is missing $\\mu\_j$
- Line 104: “denote” instead of “denotes”
- Line 130: “and the set” instead of “the set”
- Line 143: “provability” instead of “probability”
- Line 180: “Lemma” instead of “to Lemma”
- Line 187: “drawn” instead of ”drown”
- Line 188: “gets” instead of “get”
- Line 192: “contexts” instead of “context”
- In assumption 1, it could be clearer to specify that $L \\ge S$ instead of just $L>0$ for it to make sense, otherwise it might not be satisfiable
- Math display after line 195: $I \\sim p(X)$ instead of $I \\sim p(x)$
- Line 196: “that there exists an” instead of “that an”
- Line 252: the choice of naming the top-$d$ subset of pairs $(x,i)$ as $S$ might cause confusion in the reader, as $S$ is already defined to be the number $S = |\\mathcal{X}|$ of contexts

---

> ### Author Rebuttal · Authors · 2024-08-07
>
> > While the lower bound for the MwE problem improves on prior results, it requires to consider a harder setting than the standard one by restricting the learner.
>
> We deeply appreciate your suggestion.
> We agree that this is an important issue,
> and we will add this point to the abstract and introduction.
> In addition, we will mention in the revised version that
> showing a similar results in the "classical" problem setting remains as an open question.
>
>
> > For instance, the proof of Corollary 1 would benefit from a more detailed explanation of the steps; additionally, an intuitive explanation of the choice of the context-dependent learning rate would help the reader while possibly helping in adapting a similar idea in other related problem settings.
>
> The revised version will includes a step-by-step explanation of how to calculate to show the bounds in Corollary 1.
> The context-dependent learning rate is designed so that the regret upper bound of Theorem 3 is bounded well.
> In fact,
> if we consider minimizing $\sum_{x} \frac{g(x)}{\eta(x)}$ subject to the constraint of
> $\sup_{x} \frac{\eta(x)}{ g(x)^{\beta} } \le$ const,
> the optimal solution is given as $\eta(x) \propto (g(x))^{\beta}$.
>
>
> > Finally, the results in this current work still do not achieve minimax optimality...
>
> As you pointed out,
> for linear bandits and contextual linear bandits,
> minimax regret bounds have only been identified when $K$ is within a specific setting or range.
> The revised version will place more emphasis on this fact and make it clear that relaxing these assumptions is an open question to be resolved in future studies.
>
>
> > Could you expand a bit on how the argument at lines 278-279 allows us to keep the generality of the values of $S$, $K$, and $d$ as claimed?
>
> We first note Theorem 4 implies that,
> if some $d'$ satisfies $K \ge 2^{d'}$ and $d \ge d'S$,
> we can obtain a regret lower bound of $R_T = \Omega(d'\sqrt{ST})$.
> Let $(d, K, S)$ be an arbitrary given parameter set satisfying $K \le 2^d \le K^S$.
> We then have
> $\log_2 K \le d \le S \log_2 K$.
> Define $d' := \lfloor \log_2 K \rfloor$
> and $S' := \lfloor d / \log_2 K \rfloor \le S$.
> Then,
> as we have
> $K \ge 2^{d'}$ and $d \ge S' \log_2 K \ge S'd'$,
> from Theorem 4,
> we obtain a regret lower bound of
> $R_T = \Omega(d' \sqrt{S'T}) = \Omega( \sqrt{d' S' T d'} )$
> for a problem instance with parameters $(d, K, S')$.
> By combining this with
> $d' S' = \Omega( d )$
> and
> $d' = \Omega( \log K ) = \Omega( \log_+ (K \min \\{ 1, \frac{S}{d} \\}))$,
> we obtain
> $R_T = \Omega( \sqrt{
> d T \log_+ ( K \min \\{1, \frac{S}{d} \\})
> })$.
> As we have $S' \le S$,
> the same lower bound applies to the problem with parameters $(d, K, S)$ as well.
> The revised version will include a more detailed explanation like the above.
>
>
> > Do you think the ideas from the different lower bounds for linear bandits could be adapted and combined to prove a regret lower bound for arbitrary values of $K$?
>
> We believe there is hope, and we have tried that approach, but have not yet succeeded.
> What we have considered so far is a generalization based on the combination of standard multi-armed bandits ($K = d$) and hypercube bandits ($K = 2^d$).
> For example,
> we have considered a problem in which we choose one of the the $k$ hypercubes and then choose a point in that hypercube,
> i.e.,
> we set $d = d' k$ and define an action set as $\mathcal{A} = (\{ -1, 1 \}^{d'} \times \{ 0 \}^{d-d'}) \cup (
> \{ 0 \}^{d'} \times \{ -1, 1 \}^{d'} \times \{ 0 \}^{d - 2d'}) \cup \cdots \cup (\{ 0 \}^{d-d'} \times \{ -1, 1 \}^{d'})$.
> We then have $K = |\mathcal{A}| = 2^{d'} k = d 2^{d'} / d'$.
> This construction almost covers the range of $K \in [d , 2^d]$ by adjusting the parameter $d' \in [1, d]$.
> We conjecture that a lower bound of $\Omega(d'\sqrt{kT})$
> holds in this setting.
> We have been attempting to prove a lower bound using hard instances of hypercube bandits [DKH07],
> but the existing methods of analysis (e.g. [CHZ24]) do not seem directly applicable and have not yet been able to show the lower bound.
>
> > Minor comments/typos:
>
> Thank you so much for your many helpful comments.
> Each will be reflected in the revised version.
>
> Reference:
> * [DKH07] V. Dani, S. M. Kakade, and T. Hayes. The price of bandit information for online optimization. NeurIPS2007.
> * [CHZ24] H. Chen, Y. He, and C. Zhang. On interpolating experts and multi-armed bandits. ICML2024.

---

> > ### Comment · Reviewer_FSFk · 2024-08-14
> >
> > I thank the authors for the detailed response. Since my main concerns have been addressed, I am raising my score as I believe this paper provides valuable insights towards studying the minimax optimal rate of multiarmed bandit problems.

---

### Official Review · Reviewer_aY6j · 2024-07-31

**Soundness:** 3
**Presentation:** 2
**Contribution:** 3
**Rating:** 6
**Confidence:** 3

**Summary:**

This paper presents new bounds for regret minimization problem in multi-armed bandits with expert advice (MwE) and contextual linear bandits (CLB). For MwE, This paper bridges the gap between existing upper and lower bounds by establishing a new matching minimax optimal lower bound. In the case of CLBs, the authors propose a lower bound and devise an algorithm based on the follow-the-regularized-leader approach, which achieves a matching upper bound.

**Strengths:**

• Novel contribution: The paper introduces new minimax optimal lower bounds for MwE and CLB. Additionally, for CLB, it devises an algorithm that achieves a matching upper bound.
• Theoretical results: The paper provides comprehensive proofs for each of the newly introduced bounds, offering rigorous theoretical validation of the results.

**Weaknesses:**

• Motivation: The paper lacks an exploration of practical applications of the proposed work, thus indicating a deficiency in motivation. Incorporating a discussion on potential practical applications would significantly add value of the results presented in the paper.
• Future Work: The paper does not discuss potential directions for future research.
• No empirical results verifying the bounds.

I found some typos that the author(s) might want to correct.

• In line 84, ”the player chooses an expert $J_i \in [K]$ should be replaced by ”the player chooses an expert $J_t \in [N]$, correct?

• In line 99,”$j^*\in\argmin_{j\in [N]}$ is missing $\mu_j$

• In line 213: "This section provide" to "This section provides"

**Questions:**

The proposed algorithm for contextual linear bandits (CLB) requires an initial exploration policy p0 as one of its input parameters. How is this policy determined at the start of the algorithm? Are there any specific assumptions or conditions on p0 necessary for it to adhere to the presented upper bound?

**Limitations:**

The authors discuss the assumptions needed for their results to hold.

---

> ### Author Rebuttal · Authors · 2024-08-07
>
> > Motivation: The paper lacks an exploration of practical applications of the proposed work, thus indicating a deficiency in motivation. Incorporating a discussion on potential practical applications would significantly add value of the results presented in the paper.
>
> We appreciate your suggestion.
> Since our results on linear bandits and contextual linear bandits imply improvements in the regret upper bound compared to existing algorithms,
> we expect improved performance in real-world applications and believe they will be useful in practice.
> However,
> this study focuses primarily on the goal of revealing the limits of achievable regret upper bounds for the fundamental online decision-making problems,
> rather than on specific applications.
>
> > Future Work: The paper does not discuss potential directions for future research.
>
> The revised version describes open questions and future research direction more explicitly,
> as we mentioned in "Author Rebuttal" above.
>
>
> > No empirical results verifying the bounds.
>
> This paper does not include empirical results because,
> in general,
> results of some specific numerical experiments are not very informative for the purpose of validating the analysis of *minimax* regret.
> Indeed,
> the minimax regret corresponds to the *worst-case* scenario,
> i.e.,
> the problem instances that are most unfavorable loss-sequences for the algorithm.
> Numerical experiments on specific data are rarely the worst-case input,
> and it is difficult to know how close to the worst-case it is.
> The same reason may explain why many existing studies focusing on minimax regret do not include numerical experiments.
>
>
> > I found some typos that the author(s) might want to correct.
>
> We deeply appreciate your pointing out the typo.
> We consider that all of them need to be corrected as you pointed out, and we will reflect them in the revised manuscript.
>
> > The proposed algorithm for contextual linear bandits (CLB) requires an initial exploration policy p0 as one of its input parameters. How is this policy determined at the start of the algorithm? Are there any specific assumptions or conditions on p0 necessary for it to adhere to the presented upper bound?
>
> The necessary assumption on $p_0$ is described in Assumption 2 (line 196),
> where $\lambda( p_0 )$ is defined just above it.
> As long as this assumption is satisfied,
> $p_0$ can be determined in any way.
> In particular, if $p_0$ is determined using g-optimal design, then $\lambda(p_0) \le Ld$ holds,
> and hence this assumption is satisfied,
> as explained just after Assumption 2.
> The parameter $\lambda_0 = \lambda(p_0)$ depending on $p_0$ affects the asymptotically non-dominant terms (in the regime of $T \rightarrow \infty$) of the regret upper bounds provided in Theorem 3 and Corollary 1.

---

> > ### Comment · Reviewer_aY6j · 2024-08-13
> >
> > Thank you for addressing my questions.
> >
> > The phrasing of Assumption 2 is a bit confusing to me. You write "We assume that an exploration policy $p_0 : X \to \mathcal{P}(K)$ such that $\lambda(p_0) < \infty$." It feels like a word is missing. You are saying that a policy $p_0$ *$\textbf{exists}$* with $\lambda(p_0) < \infty$, correct?
> >
> > I see now why such a policy should exist. Incidentally you wrote "g-optimal" and usually I see the "G" capitalized.

---

> > > ### Author Response · Authors · 2024-08-13
> > >
> > > Thank you for your review and feedback on the manuscript and rebuttal.
> > >
> > > You are right about both of the points you raised.
> > > In the revised version, we will add the phrase "there exists" in the description of Assumption 2 and correct the "g" in "g-optimal" to the "G" capitalized.
> > >
> > > We are deeply grateful to you for bringing these errors to our attention.

---

### Author Rebuttal · Authors · 2024-08-07

# Thank you and future revisions

First of all, we would like to express our deepest gratitude to the reviewers who spent a great deal of time reviewing this paper.
Thanks to the valuable peer review comments we received,
we are confident that the quality of our manuscript will be greatly improved.
The following revisions are planned as major changes:

### Note on the setup of multi-armed bandits with expert advice
In the revised version,
the notes on problem setup explained in Remark 1 of the current manuscript will also be mentioned in the abstract and introduction.
As noted in Remark 1 and in the comment by Reviewer FSFk,
the problem setting of BwE considered in this paper is more challenging than the "classical" setting where the player can observe all expert advice before selecting an arm,
while almost all known existing algorithms,
including those in Table 1,
work for this more challenging setting.
Although these facts were written in Remark 1,
we believe that the current abstract and introduction were misleading,
as Reviewer FSFk pointed out.
To avoid such confusion, we revise the manuscript so that readers will notice this fact just by reading the intro and abstract.

### Open question and future work
In the revised version, we will add descriptions of remaining open questions and potential future research directions.
For examples:
* Tight regret bounds for "classical" setting of multi-armed bandits with expert advice
* Tight bounds for linear bandits and contextual linear bandits with *more general* parameter setting of $K$, $S$, and $d$ (please see the comment by Reviewer FSFk and our response to it)
* Relaxing the assumption on the context distribution (please see the comment by Reviewer nf4R and our response to it)

### Correction of typos and more more detailed explanations
We will correct the typos the reviewers pointed and add intuitive explanations for the unclear points you commented on, as well as more detailed explanations of the steps in the analysis.


In addition to the above,
we will revise the manuscript in response to each of the comments received.
Once again, we are deeply grateful for the tremendous amount of helpful feedback.

---

### Decision · Program_Chairs · 2024-09-25

**Decision:**

Accept (poster)

**Comment:**

This paper studies two related problems. First the authors show a novel lower bound for the setting of the CMAB problem which tightly matches the known upper upper. Second the authors propose a new algorithm for the Contextual Linear Bandit problem together with a matching lower bound.

Reviewers are mostly positive on this work, however, there are multiple important problems that need to be addressed before the final version. First the authors have to discuss the fact that the lower bound for MwE problem is in a harder setting than the standard. Next, the authors have to discuss the range of K for which their lower bounds hold for the linear bandit problem. Finally, the authors have to discuss the assumption that the proposed algorithm for the linear bandit setting requires knowledge of the contextual distribution and the computational problems in the case of an infinite number of contexts. Further, the presentation of the paper needs to be improved as multiple reviewers have pointed out. Nevertheless the works contributions are non-trivial and I am leaning towards recommending it for acceptance to the program.